# Discrete Compositional Generation via General Soft Operators and Robust Reinforcement Learning

**Marco Jiralerspong**[* 1,2], **Esther Derman**[1,2], **Danilo Vucetic**[1,2], **Nikolay Malkin**[3,4],
**Bilun Sun**[5], **Tianyu Zhang**[1,2], **Pierre-Luc Bacon**[1,2,4], **Gauthier Gidel**[1,2,4]

[1]Université de Montréal    [2]Mila    [3]University of Edinburgh    [4]CIFAR    [5]Independent

## Abstract

A major bottleneck in scientific discovery involves narrowing an exponentially large set of objects, such as proteins or molecules, to a small set of promising candidates with desirable properties. While this process can rely on expert knowledge, recent methods leverage reinforcement learning (RL) guided by a proxy reward function to enable this filtering. By employing various forms of entropy regularization, these methods aim to learn samplers that generate diverse candidates that are highly rated by the proxy function. In this work, we make two main contributions. First, we show that these methods are liable to generate overly diverse, suboptimal candidates in large search spaces. To address this issue, we introduce a novel unified operator that combines several regularized RL operators into a general framework that better targets peakier sampling distributions. Second, we offer a new, robust RL perspective of this filtering process. The regularization can be interpreted as robustness to a compositional form of uncertainty in the proxy function (i.e., the true evaluation of a candidate differs from the proxy's evaluation). Our analysis leads us to a novel, easy-to-use algorithm we name trajectory general mellowmax (TGM): we show it identifies higher quality, diverse candidates than baselines in both synthetic and real-world tasks. Code: `https://github.com/marcojira/tgm`.

## 1 Introduction

The task of scientific discovery centers on discovering novel objects with desirable properties such as antimicrobial resistance or binding affinity. These objects are often discrete and structured such that they can be constructed through a sequence of compositional steps, e.g., proteins as sequences of amino acids, molecules composed of various smaller fragments (Angermueller et al., 2019; Wang et al., 2023). Due to this compositionality, the number of possible objects grows exponentially with object size and becomes too large to search exhaustively. Additionally, experimentally evaluating potential objects can be too technical or too expensive. Nonetheless, in many cases, researchers have access to a proxy reward model that quantitatively approximates the satisfaction level of the property of interest (Davis et al., 2018; Du et al., 2022).

A large body of work leverages this proxy reward function to narrow the set of objects to a small subset of promising candidates to test in a lab (Currin et al., 2015; Gubernatis & Lookman, 2018; Wu et al., 2021; AI4Science et al., 2023). An alternative approach that has demonstrated growing popularity and effectiveness is to pose this problem as an RL task (De Cao & Kipf, 2018; Darvariu et al., 2021; Bou et al., 2024). In what we will denominate discrete compositional processes (DCP), object construction is viewed as a Markov decision process (MDP) where terminal states correspond to complete objects and yield a reward given by the evaluation of that object by the proxy reward function. Then, the learned policy can be used as a generative model that samples promising candidates.

Among these approaches, generative flow networks (GFNs) have shown considerable success (Bengio et al., 2021; Jain et al., 2023; Zhou et al., 2024; Cipcigan et al., 2024). GFNs learn an amortized policy and, at optimality, sample objects with probability proportional to the exponential of their reward

---

[*]Correspondence to `marco.jiralerspong@mila.quebec`

(Eq. 2). Connections with maximum-entropy RL have been established, showing that this objective underpins the diversity of generated samples (Tiapkin et al., 2024; Mohammadpour et al., 2024a; Deleu et al., 2024). While this exact sampling goal is useful in certain applications, it is problematic in exponentially large sets: the probability of sampling from a small set of objects with high rewards is dominated by the probability of sampling from an exponentially large set of objects with low rewards. In this work, we leverage the connection between GFNs and regularized RL to use soft RL operators for scientific discovery. By combining these approaches, we derive an improved operator for the task and offer a new perspective on regularization. Our contributions can be summarized as follows.

**Our contributions:** (1) We introduce **general mellowmax**, an interpolating operator that allows for a more flexible tradeoff between quality and diversity than previous operators; (2) We derive a trajectory constraint for this operator, which we refer to as **trajectory general mellowmax** (TGM). TGM enables us to propose a GFN-like algorithm for scientific discovery tasks that empirically finds diverse modes with higher reward than soft mellowmax, GFNs, and other RL baselines; (3) We provide a robust RL interpretation of these baselines in DCPs. Specifically, we show that the reward uncertainty sets induced by TGM are more interpretable than those associated with entropy regularization or GFNs. The analytical techniques we employ pave the way to more general trajectory constraints through an interpretable equivalence between regularization and reward robust sets.

## 2 RELATED WORK

The goal of sampling proportional to reward targeted by GFNs was historically tackled using Markov chain Monte Carlo (MCMC) (Gilks et al., 1995). These methods construct chains through a random walk: from a set of initial points, new points are proposed and either accepted or rejected based on their function value (here, their proxy reward). While theoretically appealing, these methods struggle in high dimensions, in particular when it comes to discovering separate modes (Bengio et al., 2021).

On the other hand, GFNs have demonstrated success in protein design (Jain et al., 2022), drug discovery (Shen et al., 2023), and material design (AI4Science et al., 2023). The standard way to tackle the excessive smoothness of GFN policies is through temperature conditioning (Zhang et al., 2023; Kim et al., 2023; Zhou et al., 2024): the GFN target can be posed as learning $p(x) \propto e^{\beta r(x)}$ where $\beta$ is an inverse temperature parameter. Temperature conditioning involves learning a policy conditionally on $\beta$, with $\beta$ varying throughout training. Another successful approach involves simultaneously learning a Q-function and sampling using a mixture of the GFN policy and the argmax over Q-values (Lau et al., 2024). Both methods complement our approach, as they can be used in addition to our framework. Mohammadpour et al. (2024b) have also proposed a dynamic regularization coefficient based on the number of actions, for maximum entropy RL. This change can also be used to address excessive smoothness in GFNs or incorporated through a varying $\omega_s$ in our Eq. 7.

As for the RL perspective, many soft RL methods fit in the regularized framework of Geist et al. (2019). Other forms of regularization and their corresponding operators include the mellowmax operator (Asadi & Littman, 2017), which takes the logmeanexp of Q-values instead of the logsumexp, and its extension, the soft mellowmax operator (Gan et al., 2021). Jiralerspong et al. (2024) also combine a GFN perspective with RL by developing an adversarial version of the mellowmax operator called AFlowNets. None of these works use their operators for DCPs, whereas we generalize them, integrate them with trajectory constraints, and successfully apply these constraints to real DCP tasks.

Orthogonally, one can take a generative modeling (with a focus on property optimization) approach to the problem of scientific discovery. Of particular interest are methods that take a pretrained diffusion model and seek to finetune it to produce samples with a given property (in our case, the output of the proxy reward function) while maintaining high likelihood under the base model. Uehara et al. (2024a;b) do so for continuous diffusion models while Wang et al. (2024) uses the gumbel softmax trick to differentiate with respect to a discrete diffusion model. Interestingly, these works also use an entropy term to ensure diversity of generated samples. While this approach is also applicable to similar problems, it additionally requires a pretrained diffusion model and can be computationally expensive (for example due to the need to approximately solve an SDE at every step of training).

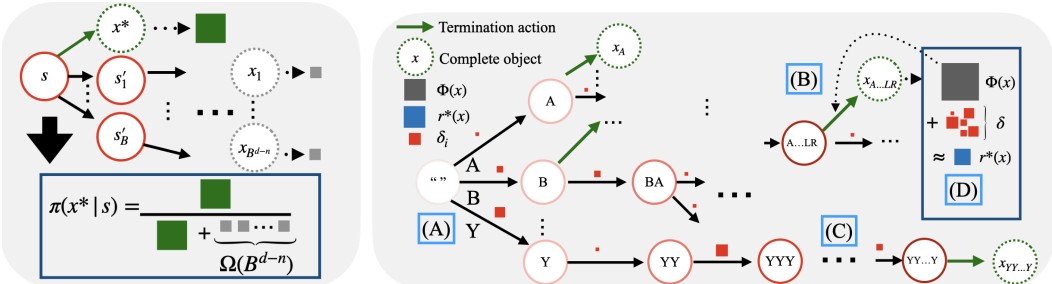

Figure 1: (Left) Illustrated issue with sampling proportional to reward. The high reward of the optimal sequence (green) is drowned out by all the rewards in the combinatorial explosion of lower-reward longer subsequences. (Right) A DCP for a protein sequence generation task. (A) Starting from an empty string, amino acids are added sequentially until termination. (B) Then, the full sequence is evaluated by a proxy reward function $\Phi(x)$ whose value is given as reward for the termination action. (C) Over the course of protein generation, uncertainty accumulates through the per-step uncertainty terms $\delta_i$. (D) The true reward depends on $\Phi(x)$ *and the accumulated uncertainty.*

## 3 BACKGROUND AND MOTIVATION

**MDPs.** An MDP is a tuple $(\mathcal{S}, \mathcal{A}, \gamma, P, r)$ where $\mathcal{S}$ and $\mathcal{A}$ are finite state and action spaces respectively, $\gamma \in [0, 1)$ is a discount factor, $P : \mathcal{S} \times \mathcal{A} \to \Delta_{\mathcal{S}}$ a transition kernel and $r : \mathcal{S} \times \mathcal{A} \to \mathbb{R}$ a reward function. Define $\Pi$ as the set of policy mappings $\pi : \mathcal{S} \to \Delta_{\mathcal{A}}$. The goal is to find $\pi \in \Pi$ that maximizes the value function $v_r^{\pi}(s) := \mathbb{E}^{\pi}[\sum_{t=0}^{\infty} \gamma^t r(s_t, a_t) \mid s_0 = s]$, $\forall s \in \mathcal{S}$. The dependence of the value function on the reward function is made explicit through the subscript $r$, as clarified in the next paragraph. For any policy $\pi \in \Pi$, the expected reward is $r^{\pi}(s) := \sum_{a \in \mathcal{A}} \pi_s(a) r(s, a)$ and the expected transition is $P^{\pi}(s, s') := \sum_{a \in \mathcal{A}} \pi_s(a) P(s, a, s')$. The evaluation Bellman operator given by $T_r^{\pi} v := r^{\pi} + \gamma P^{\pi} v$ for all $v \in \mathbb{R}^{\mathcal{S}}$ is known to be contracting, admitting $v_r^{\pi}$ as a fixed point. Similarly, the optimal value function $v_r^*(s) := \max_{\pi \in \Pi} v_r^{\pi}(s)$ is the fixed point of the optimal Bellman operator $T_r^* v(s) := \max_{\pi \in \Pi} T_r^{\pi} v(s), \forall v \in \mathbb{R}^{\mathcal{S}}, s \in \mathcal{S}$. An optimal policy can be derived from iterative Bellman updates, which form the building block of RL (Puterman, 2014).

**Robust MDPs.** The true reward or transition function of an MDP is rarely known in practice. It may be estimated from trajectory data, but a small error on the MDP model can alter policy performance (Mannor et al., 2004). The robust MDP setting addresses this issue by assuming that $(P, r)$ is unknown, lying in a given uncertainty set. In our DCP setting where transitions are fully determined by the performed actions, we reasonably assume that the reward model $r \in \mathcal{R}$ is the only uncertain element. Then, we aim to maximize performance for the worst-case model, namely:

$$v_{\mathcal{R}}^{\pi}(s) := \min_{r \in \mathcal{R}} v_r^{\pi}(s), \qquad \forall s \in \mathcal{S}.$$

To solve this max-min problem, one can resort to robust Bellman operators $T_{\mathcal{R}}^{\pi} v := \min_{r \in \mathcal{R}} T_r^{\pi} v$ and $T_{\mathcal{R}}^* v := \max_{\pi \in \Pi} T_{\mathcal{R}}^{\pi} v$, $\forall v \in \mathbb{R}^{\mathcal{S}}$. Indeed, both are contracting and admit the robust value $v_{\mathcal{R}}^{\pi}$ and the optimal robust value $v_{\mathcal{R}}^*$ as fixed points, respectively (Iyengar, 2005; Wiesemann et al., 2013).

**Regularized MDPs** provide a general framework for regularization in RL, recovering celebrated algorithms such as soft Q-learning (Haarnoja et al., 2017). A regularized MDP is an MDP $(\mathcal{S}, \mathcal{A}, \gamma, P, r)$ combined with a family $\Omega := (\Omega_s)_{s \in \mathcal{S}}$ of convex functions $\Omega_s : \Delta_{\mathcal{A}} \to \mathbb{R}$. At each state $s \in \mathcal{S}$, $\Omega_s$ defines a policy regularizer $\Omega_s(\pi_s)$, for $\pi_s \in \Delta_{\mathcal{A}}$. The regularized Bellman operator is given by:

$$[T^{\pi, \Omega} v](s) := T_r^{\pi} v(s) - \Omega_s(\pi_s), \qquad \forall v \in \mathbb{R}^{\mathcal{S}}, s \in \mathcal{S}.$$

and its greedy equivalent by $[T^{*, \Omega} v](s) := \max_{\pi_s \in \Delta_{\mathcal{A}}} [T^{\pi, \Omega} v](s)$ (Geist et al., 2019). Derman et al. (2021) have established an equivalence between policy-regularized and robust Bellman operators, thus highlighting a formal motivation for regularized RL. The statement below is a direct reformulation of (Derman et al., 2021, Thm. 3.1).

**Theorem 3.1** (Derman et al. (2021)). *Assume that the reward function $r$ is uncertain and satisfies $r_s \in \mathcal{R}_s := r_0(s, \cdot) + \tilde{\mathcal{R}}_s$, where $\tilde{\mathcal{R}}_s \subseteq [-R, R]^{\mathcal{A}}$ is closed and convex for all $s \in \mathcal{S}$. Then, for any $\pi \in \Pi$, the robust value function $v_{\mathcal{R}}^{\pi}$ is the fixed point of the regularized Bellman operator $T^{\pi, \Omega}$ with $\Omega_s(\pi_s) := \max_{r_s \in \tilde{\mathcal{R}}_s} \langle -\pi_s, r_s \rangle$. In other words, it holds that $v_{\mathcal{R}}^{\pi}(s) = T_{r_0}^{\pi} v_{\mathcal{R}}^{\pi}(s) - \Omega_s(\pi_s), \forall s \in \mathcal{S}.$*

**Discrete compositional processes (DCP)** are special instances of standard MDPs with deterministic transitions. They can fully be described by a tuple $(\mathcal{G}, \Phi)$, where $\mathcal{G}$ is a directed acyclic graph with a single source state $s_0$ and a set of terminal states $\mathcal{X}$ (see Fig. 1). Each node in the graph corresponds

Figure 2: Illustration of empirical distribution of proxy rewards for 3 domains compared to rewards of objects sampled by a TGM policy at the end of training (blue). The uniform histogram (grey) was generated by sampling and evaluating 1 million uniformly randomly drawn samples for each task, showing that the vast majority of possible objects are assigned a non-negligible reward by the proxy reward function. Max score refers to the highest proxy reward given to an object in the validation set (see App. F.4 for a description of the validation set).

to a partial object, $s_0$ being the empty set. An edge $(s \to s')$ represents either adding a part to the object $s$ to get $s'$ or a termination action if $s' \in \mathcal{X}$. Each completed object $x \in \mathcal{X}$ has an associated reward $\Phi(x)$ given by the proxy reward model $\Phi : \mathcal{X} \to \mathbb{R}$.

In standard RL, the goal is to maximize the discounted cumulative reward. Instead, for scientific discovery, we are interested in finding the most promising set of $k$ candidates that are sufficiently distinct from one another (Jain et al., 2022). In practice, an imperfect but computationally efficient proxy score $\Phi$ is used to estimate the usefulness of each candidate, so naively maximizing usefulness can be problematic. Instead, given some metric $d$ between completed objects, practitioners approximately aim to find a diverse and novel (w.r.t. the training set) set of candidates $\{x_1, \ldots x_k\}$ with a high score (Jain et al., 2022). This goal can be formalized as maximizing the *average mode reward* (for simplicity, we only formalize the diversity constraint):

$$\max_{\{x_1, \ldots x_k\} \subset \mathcal{X}} \frac{1}{k} \sum_{i=1}^{k} e^{\Phi(x_i)} \quad \text{subject to} \quad d(x_i, x_j) > d_{\min}, \quad \forall i \neq j. \tag{1}$$

### 3.1 LIMITATIONS OF GFLOWNETS IN SCIENTIFIC DISCOVERY

Under a GFN perspective, the exponential of the proxy reward function can be viewed as an unnormalized probability mass. Then, the goal is to learn a per-state policy $\pi_s$ whose sequential application samples objects proportionally to this function. Formally, if we denote by $\mathcal{T}(x)$ all trajectories in $\mathcal{G}$ ending at terminal state $x \in \mathcal{X}$ and $\pi_{\tau(x)}$ the probability of picking all edges from a trajectory $\tau(x)$ under policy $\pi$, the objective is to match the following distribution:

$$p(x) := \sum_{\tau(x) \in \mathcal{T}(x)} \pi_{\tau(x)} \propto e^{\beta \Phi(x)}, \qquad \forall x \in \mathcal{X}. \tag{2}$$

where $\beta$ is the inverse temperature. We argue this objective is suboptimal for solving Eq. 1.

**Motivating example.** The main issue with Eq. 2 is that it yields a distribution assigning higher probability to a large number of suboptimal reward objects rather than to a small number of high-reward objects. To demonstrate this, consider a DCP (illustrated in Fig. 1) where the goal is to generate sequences of maximum length $d$ from a vocabulary of tokens of size $B$. Suppose a sampler is at an optimal sequence $s^*$ of length $n < d$. The sampler's policy then needs to decide whether to terminate and return $x^*$, or to add more tokens and harm the usefulness of the sequence. In doing so, it weighs $e^{\Phi(x^*)}$ against the reward of all sequences starting with $s$. There are $\Omega(B^{d-n})$ such sequences. Suppose the sampler perfectly matches (2) and these sequences have rewards lower bounded by $c_r > 0$. Then, $\pi(x^* \mid s) \leq e^{\Phi(x^*)} / r \cdot B^{d-n}$, and the probability of returning the optimal sequence at $s$ decreases exponentially in $d - n$. The problem occurs in many applications, as *almost all objects have non-negligible reward* (see Fig. 2). When these empirical distributions are representative, GFNs may conservatively estimate the probability of sampling a promising candidate.

## 4 ALTERNATIVE OPERATORS

To address this limitation, we propose taking a regularized RL perspective and modifying the operator used by GFNs. To simplify the discussion, we examine the case where $\mathcal{G}$ in the DCP is a tree. In such settings, there is a direct equivalence between the GFN flow operator and the soft Bellman operator

|  | **GFN** | **Soft Bellman** | **MM** | **SMM** | **GM (ours)** |
|---|---|---|---|---|---|
| **Accumulation** | $\log(k)$ | $\frac{\log(k)}{\omega}$ | $0$ | $0$ | $\frac{(1-\mathsf{q})\log(k)}{\omega}$ |
| **Dilution** | $0$ | $0$ | $\frac{\log(k)}{\omega}$ | $\frac{\log(k)}{\alpha+\omega}$ | $\frac{\mathsf{q}\log(k)}{\mathsf{q}\alpha+\omega}$ |
| $\|\Delta_s^T\|$ | $\log(k)$ | $\frac{\log(k)}{\omega}$ | $\frac{\log(k)}{\omega}$ | $\frac{\log(k)}{\alpha+\omega}$ | $\max\left(\frac{(1-\mathsf{q})\log(k)}{\omega}, \frac{\mathsf{q}\log(k)}{\mathsf{q}\alpha+\omega}\right)$ |

Table 1: Worst-case accumulation, dilution and $|\Delta_s^T|$ for different operators when $\frac{\mathsf{q}\alpha+\omega}{\alpha} > 1$. $k$ is the number of actions at a given state, $\omega$ is the regularization coefficient, $\alpha$ is the softmax inverse temperature and $\mathsf{q}$ is the GM interpolation factor. For the same values of $\alpha, \omega, k$, general mellowmax with $\mathsf{q} \in (0,1)$ trades off accumulation and dilution to get a lower worst-case $|\Delta_s^T|$ than the other operators. See App. C for proofs.

with temperature $\omega = 1$. Denoting $Q_s(a) := v_{s_a} + r(s,a)$ (with $s_a$ being the state reached by taking action $a$ from $s$) and the vector of Q-values at a state by $\mathbf{Q_s}$, the soft Bellman operator is given by:

$$T^{\text{SB}}v_s := \frac{1}{\omega}\text{logsumexp}(\omega\mathbf{Q_s}). \tag{3}$$

Recursively applying this operator in a DCP, going from the leaves of $\mathcal{G}$ to the root, causes value to accumulate. At optimality, the value of a state is the *logsumexp* of the rewards of all objects that can be reached from that state. The accumulation of the value of a large number of suboptimal objects can overwhelm one high-reward object.

Instead of taking the sum, we could take the *logmeanexp* of the Q-values. Doing so yields the mellowmax operator (Asadi & Littman, 2017). While the mellowmax operator solves the accumulation issue, it is liable to the opposite issue, dilution. Due to this averaging, the high reward of an object deep in the tree can get diluted by lower reward objects as we approach the root (since we divide by the number of children at each state).

$$T^{\text{MM}}v_s := \frac{1}{\omega}\log\sum_{a \in \text{Ch}(s)}\frac{1}{|\text{Ch}(s)|}e^{\omega Q_s(a)}. \tag{4}$$

Luckily, we can alleviate the dilution issue by weighting the elements of the sum proportionally to their value. In particular, we can use the softmax (controlled by $\alpha$) of the Q-values as weights (instead of an equal weighting) and thus obtain the soft mellowmax operator (Gan et al., 2021):

$$T^{\text{SMM}}v_s = \frac{1}{\omega}\log\langle\text{softmax}(\alpha\mathbf{Q_s}), e^{\omega\mathbf{Q_s}}\rangle. \tag{5}$$

Both accumulation and dilution consist of deviations between the application of the operator and the max Q-value: $\Delta_s^T := Tv_s - \max_{a \in \text{Ch}(s)} Q(s,a)$. Formally, we define them as follows:

$$\text{Acc}^T(s) := \max(\Delta_s^T, 0) \quad \text{Dil}^T(s) := \max(-\Delta_s^T, 0). \tag{6}$$

Respectively, they represent the increase or decrease in the value of a state relative to its maximum Q-value. Ultimately, for scientific discovery applications, the ideal operator/associated policy would assign probability to all actions *while* being resistant to accumulation and dilution.

We propose a new operator that interpolates between the accumulation of the soft Bellman operator and the dilution of the soft mellowmax operator with a hyperparameter $\mathsf{q}$. Values of $\mathsf{q}$ in $(0,1)$ have a smaller amount of dilution and accumulation instead of a large amount of either. Consequently, the resulting operator trades off both issues and has better worst-case bounds on $|\Delta_s^T| = \max\{\text{Acc}^T(s), \text{Dil}^T(s)\}$, as illustrated in Tab. 1.

### 4.1 GENERAL MELLOWMAX

The operators mentioned above can be viewed through the lens of regularized MDPs. On one hand, the soft Bellman operator corresponds to regularizing by the Shannon entropy. On the other hand, given a distribution $d_s \in \Delta_{\mathcal{A}}$, the mellowmax/soft mellowmax operators result from regularization by $\text{KL}(\pi_s, d_s)$ where $d_s$ is the uniform distribution or $\text{softmax}(\alpha\mathbf{Q_s})$ respectively.

We define the **general mellowmax (GM)** regularizer by interpolating between these regularizers. Then, we can recover these operators as special cases and also trade off their respective effects. We denote the general regularizer by $\Omega^{\mathsf{q}}_{d_s}$ and the case $d_s = \mathrm{softmax}(\alpha \mathbf{Q_s})$ by $\Omega^{\mathsf{q}}_\alpha$.

$$\Omega^{\mathsf{q}}_{d_s}(\pi_s) := \frac{1}{\omega}\left[\mathsf{q}\mathrm{KL}(\pi_s, d_s) + (1-\mathsf{q})(-\mathrm{H}(\pi_s))\right], \quad \forall s \in \mathcal{S}, \pi_s \in \Delta_\mathcal{A}, \omega \in \mathbb{R}_{>0}, \mathsf{q} \in [0,1]. \quad (7)$$

Notably, unlike other regularizers, $\Omega^{\mathsf{q}}_\alpha$ depends on the Q-values of the current state. Consequently, the associated operator is not a convex conjugate and deviates from the framework of (Geist et al., 2019). Nonetheless, we can still derive the associated operator which we denote $\Omega^{\mathsf{q},*}_\alpha$ using similar techniques (see App. E):

$$\Omega^{\mathsf{q},*}_\alpha(\mathbf{Q_s}) = \frac{1}{\omega}\log\langle\mathrm{softmax}(\alpha\mathbf{Q_s})^{\mathsf{q}}, e^{\omega Q_s}\rangle. \quad (8)$$

Using the regularized interpretation, $\mathsf{q}$ trades off between maximizing the policy's entropy and having a policy that is close to the softmax of the Q-values. $\alpha$ allows us to put more or less weight on actions with high Q-values while $\omega$ controls the weight of regularization. In particular, setting $\mathsf{q} = 0$ allows us to recover the standard entropy-regularized / GFN operators and policy. On the other hand, by setting $\mathsf{q} = 1$, we recover the soft mellowmax operator (with $\alpha = 0$ we recover the mellowmax operator). The interpolated operator captures three important soft RL operators and allows us to smoothly control the tension between accumulation/dilution using $\mathsf{q}$ and $\alpha$.

We also observe that $\Omega^{\mathsf{q}}_{d_s}$ is equivalent to a KL between a policy and a tilted softmax, as stated below.

**Proposition 4.1.** *For all* $\mathsf{q} \in [0,1]$, $\Omega^{\mathsf{q}}_{d_s}(\pi_s) = \frac{1}{\omega}\mathrm{KL}(\pi_s, d_s^{(\mathsf{q})}) - \frac{1}{\omega}\log(Z_{\mathsf{q}}(d_s))$, *where* $Z_{\mathsf{q}}(d_s) :=$ $\sum_{a \in \mathcal{A}} d_s(a)^{\mathsf{q}}$ *and* $d_s^{(\mathsf{q})} = d_s^{\mathsf{q}}/Z_{\mathsf{q}}(d_s)$ *is the* $\mathsf{q}$-*tilted softmax distribution.*

## 4.2 TRAJECTORY GENERAL MELLOWMAX

Instead of training on transitions, it is possible to leverage trajectory-level constraints that relate policy/value functions over multiple steps to recover training algorithms that use subtrajectories as data units. In soft RL, this connection was first formalized in Nachum et al. (2017) through path consistency learning (PCL) for maximum entropy RL. Path consistency objectives have since been derived for Tsallis entropy (Chow et al., 2018) as well as the general class of $\alpha$-divergences (Brekelmans et al., 2022).

Trajectory-level constraints have been shown to be crucial to the performance of methods in DCP tasks. Trajectory balance (TB) (Malkin et al., 2022) and subtrajectory balance (Madan et al., 2023) are consistently used in GFNs, as DCP tasks only have terminal rewards. By connecting the policy at early states to this terminal reward, these constraints are noticeably better at propagating signal and thus improving credit assignment.

**Theorem 4.1** (Trajectory GM). *For any DCP* $(\mathcal{G}, \Phi)$, *let* $(\pi^*, Q^*)$ *be the unique optimal value/policy functions for* $\Omega^{\mathsf{q},*}_\alpha$. *Then, for a given Q-function* $Q^\theta$, *the two following statements are equivalent:*

$$\mathrm{softmax}([\alpha\mathsf{q} + \omega]Q^\theta_s) = \arg\max_{\pi_s}\langle\pi_s, Q^*_s\rangle - \frac{1}{\omega}\Omega^{\mathsf{q}}_\alpha(\pi_s), \text{ for all states } s \in \mathcal{G}, \quad (9)$$

*if and only if*

$$\left. v_0^* + \sum_{i=0}^n \frac{1}{\omega}\left(\log\mathrm{softmax}([\alpha\mathsf{q} + \omega]Q^\theta_{s_i})[a_i] - \mathsf{q}\log\mathrm{softmax}(\alpha Q^\theta_{s_i})[a_i]\right) = r(x), \atop \text{for all trajectories } s_0 \xrightarrow{a_0} \cdots \xrightarrow{a_{n-1}} s_n \xrightarrow{a_n} x \text{ in the DCP.} \right\} \quad (10)$$

We derive a novel, equivalent trajectory constraint for the general mellowmax operator. In particular, Thm 4.1 makes explicit that satisfying the trajectory constraint Eq. 10 on all trajectories yields $Q_\theta$ such that $\mathrm{softmax}([\alpha\mathsf{q} + \omega]Q_\theta)$ is the optimal policy for the regularized problem. See App. E.3 for the proof.

**Trajectory general mellowmax (TGM).** We now instantiate the interpolated regularizer as a practical algorithm for use in DCPs. Specifically, we make three design decisions.

(1) Using the general mellowmax operator with $d_s = \text{softmax}(\alpha Q_s)$. Doing so allows us to use TGM both on-policy and off-policy and does not require a separate $d_s$. Instead, the policy seeks to maximize reward while being close to a softmax of the learned Q-values.
(2) Aiming to satisfy a novel trajectory constraint derived in App. E.3. Doing so recovers the benefits of trajectory constraints for DCPs.
(3) Training a single network $Q^\theta$ of Q-values through the VarGrad objective of Richter et al. (2020); Zhang et al. (2023). It can be shown that minimizing the trajectory constraint Eq. 10 is equivalent to minimizing the following loss (where $\sigma_t$ denotes the softmax with inverse temperature $t$):

$$\mathcal{L}_{\text{TGM}, Q_\theta}(\tau) = \text{Var}\left[\frac{1}{\omega}\left(\sum_{i=0}^{n}\log\sigma_{\mathsf{q}\alpha+\omega}[Q_{s_i}^\theta(a_i)] - \mathsf{q}\log\sigma_\alpha[Q_{s_i}^\theta(a_i)]\right) - \beta r(x)\right].$$

From $Q^\theta$, the optimal policy is easily computed. The resulting algorithm is easy to implement, efficient (particularly for transformers), and is equivalent to GFN training when $q = 0, \omega = 1$.

## 5 DISCRETE COMPOSITIONAL GENERATION VIA ROBUST RL

In this section, we offer a robust RL interpretation of regularized operators for DCPs. As described in Thm. 3.1, regularized MDPs are known to be equivalent to robust MDPs with uncertain reward. In the context of scientific discovery, since the reward is the evaluation of objects by a proxy function, we can conceptually consider the existence of a *hidden true reward* $r^*$ that we would like to maximize[1]. The filtering process can be interpreted as an attempt to robustly maximize the proxy reward $\Phi$, accounting for the difference $\delta$ between $\Phi$ and $r^*$. Thus, the goal is to solve:

$$\max_{p \in \Delta_\mathcal{X}} \min_{\delta \in \mathcal{R}} \mathbb{E}_{x \sim p}[\Phi(x) + \delta(x)] \tag{11}$$

where $\mathcal{R} \in \mathbb{R}^{|\mathcal{X}|}$ is the uncertainty set. The question then remains of how this uncertainty should be modeled in $\mathcal{G}$. Traditionally, the reward in a DCP is identically 0 everywhere except for terminating actions. However, assuming that uncertainty only exists at the last action misses the compositional nature of the task. Instead, we decompose $\delta(x)$ into a sum of perturbations occurring at each step of the generation process. More precisely, given a trajectory $s_0 \xrightarrow{a_0} \cdots \xrightarrow{a_{n-1}} s_n = x$, we have

$$\delta(x) = \sum_{i=0}^{n} \delta_i[a_i], \quad \text{where } \delta_i \in \mathcal{R}_{s_i}. \tag{12}$$

In this formulation, the uncertainty on $\Phi$ is *split between all actions taken to construct the object*, instead of only existing at the final action, as illustrated in Fig. 1. From a robust RL perspective, this uncertainty set structure corresponds to the common assumption of state-rectangularity (Wiesemann et al., 2013; Gadot et al., 2024). The following section makes this model of uncertainty more explicit and analyzes the uncertainty sets of different operators. In particular, we show that the reward uncertainty set entailed by GFNs is inadequate, whereas the set induced by the soft mellowmax operator provides a more meaningful notion of uncertainty.

### 5.1 FENCHEL-ROBUST FORMULATION OF REGULARIZED MDPS

In this subsection, we use the robust MDP notations introduced in § 3. The following result provides an explicit mapping between reward-robust MDPs, as defined in (12), and regularized MDPs, through equivalent value functions.

**Theorem 5.1** (Fenchel-Robust MDP). *For any state $s \in \mathcal{S}$, let $\Omega_s : \Delta_\mathcal{A} \to \bar{\mathbb{R}}$ be a proper convex regularization function. Denote by $\Omega_s^*$ its convex conjugate which is known to be proper and convex. For any $s \in \mathcal{S}$, define the reward set*

$$\mathcal{R}_s := r_0(s, \cdot) + \left\{r_s \in \mathbb{R}^\mathcal{A} : \Omega_s^*(-r_s) \leq 0\right\}.$$

*Then, denoting the robust operator for $\mathcal{R}_s$ by $T_{\mathcal{R}_s}^\pi$ and the regularized operator for $\Omega_s$ by $T^{\pi,\Omega_s}$:*

$$T_{\mathcal{R}_s}^\pi v = T^{\pi,\Omega_s} v, \qquad \forall \pi \in \Pi, v \in \mathbb{R}^\mathcal{S}.$$

---

[1]This reward would correspond to the effective properties of interest measured in a lab.

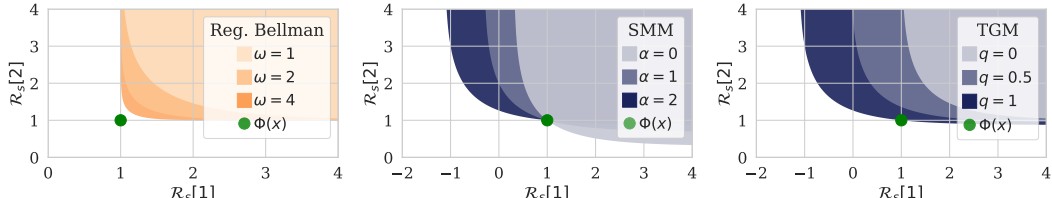

Figure 3: (Left) Regardless of $\omega$, the uncertainty set *never contains* $\boldsymbol{\Phi}(\mathbf{x})$. As a result, the soft Bellman/GFN operator is only robust to increases in reward. (Middle) For the soft mellowmax operator, for different values of $\alpha$ with $d_s[1] > d_s[2]$, the uncertainty set contains $\boldsymbol{\Phi}(\mathbf{x})$. Thus, the operator is robust to decreases in reward of one object (but not both at the same time). When $\alpha = 0$ (mellowmax), there is a symmetry in this tradeoff, while increasing $\alpha$ skews it such that the object with higher $d_s$ only admits a small decrease in reward. (Right) The uncertainty sets for GM interpolate between the two effects. While the uncertainty set for $0 < \mathsf{q} < 1$ does not contain $\boldsymbol{\Phi}(\mathbf{x})$, it contains points corresponding to decreases in reward.

Based on this theorem, solving Eq. 11 is equivalent to solving the associated regularized MDP. Thm. 5.1 uses convex conjugacy in the context of dynamic programming. Similar results can be found in previous works (Eysenbach & Levine, 2021; Husain et al., 2021; Brekelmans et al., 2022; Derman et al., 2021), but as we carefully detail in App. A.5, they do not directly apply to our setting.

## 5.2 Robust sets induced by common regularizers

Given this perspective, we now analyze the uncertainty sets corresponding to the regularizers we considered above. Since each is a special case of GM, we first give the uncertainty set of GM:

$$\mathcal{R}_s := r_0(s, \cdot) + \left\{ r_s \in \mathbb{R}^{\mathcal{A}} : \tfrac{1}{\omega} \sum_{a \in \mathcal{A}} d_s(a)^{\mathsf{q}} e^{-\omega r_s(a)} \le 1 \right\}. \tag{13}$$

A proof is in App. A.4. As opposed to Derman et al. (2021), the fact that our uncertainty sets are state-rectangular makes them independent of the executed policy, which results in a formulation that fits with the standard robust MDP setting (Wiesemann et al., 2013).

Fig. 3 shows the uncertainty sets induced by three regularizers: negative Shannon entropy, soft mellowmax, and GM. For this illustration, we consider a single-step generation process with 2 possible objects and associated proxy rewards $[1, 1]$. Similarly to (Brekelmans et al., 2022), we find that the entropy regularized Bellman operator (left) fails to capture a meaningful notion of robustness. Indeed, the uncertainty set never contains the proxy $\Phi(x)$, although higher values of $\omega$ do bring the set closer to the proxy reward. This problem is significantly exacerbated in a DCP with multiple steps: at each layer, the uncertainty set becomes further away from the proxy reward (see App. D).

## 6 Experiments

Our experiments aim to answer three main questions:

**[Q1]** How does the parameter q affect the peakiness of sampling in small and synthetic environments?
**[Q2]** Does TGM find better candidates than standard methods in large biological design DCP tasks?
**[Q3]** How robust is TGM to changes in hyperparameter settings?

### 6.1 Impact of q in small and synthetic environments [Q1]

**TF-Bind-8.** Originally proposed in (Trabucco et al., 2022), TF-Bind-8 generates DNA sequences of length 8 to find those with high binding activity to human transcription factors. The reward is the experimental binding activity of the sequence from (Barrera et al., 2016). As there are only $4^8$ such sequences, the search space is small enough to compute the optimal value for each operator. We compare the optimal sampling densities of each algorithm in Fig. 4 for $\beta = 4$, $\alpha = 2$ and $\omega = 2$.

**Bit sequence.** Proposed in (Malkin et al., 2022), the bit sequence task is significantly larger with $2^{120}$ possible sequences. The goal is to generate bit sequences of length $n$ by adding $k$ bits at a time. $M$ modes are selected semi-randomly and the reward is given by $r(x) = 1 - \min_{y \in M} d(x, y)/n$ where $d$ is the Levenshtein edit distance (Levenshtein et al., 1966). Loosely, the reward of a sequence is the negative normalized edit distance to the nearest mode.

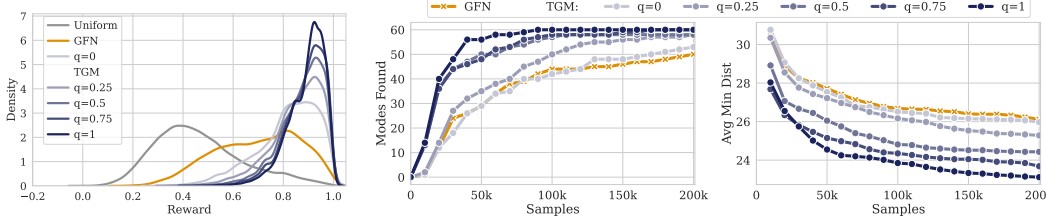

Figure 4: (Left) Comparison of the optimal sampling distribution of GFN and variants of TGM for TF-Bind-8 rewards. For the same $\beta = 4$, TGM concentrates significantly more mass on the upper quantiles of the reward distribution. (Middle) Number of modes found by TGM and GFN. The increased peakiness of the TGM sampling does not harm its ability to find different modes. (Right) Average (over modes) of the distance of the closest sample found for each mode. On average, increasing q allows TGM to find closer samples to the true modes.

|  | SAC | PPO | GFN | TGM (ours) | | | | | Valid. |
|  |  |  |  | q = 0 | q = 0.25 | q = 0.5 | q = 0.75 | q = 1.0 | Max |
|---|---|---|---|---|---|---|---|---|---|
| **UTR** | 3.82±.03 | 3.46±.02 | 4.12±.00 | 4.19±.01 | 4.15±.01 | 4.25±.00 | **4.27±.01** | 4.17±.01 | 4.26 |
| **AMP** | 5.69±.35 | 9.79±.02 | 10.04±.04 | 10.13±.04 | 9.93±.03 | 9.82±.02 | **10.43±.07** | 10.29±.05 | 9.96 |
| **GFP** | — | 0.66±.02 | 1.90±.03 | 2.44±.09 | **2.95±.08** | 2.66±.12 | 2.34±.11 | 2.05±.11 | 3.63 |

Table 2: Average mode reward of TGM compared to baselines on the biological sequence design tasks. We report the average and standard error over 15 seeds. In all domains, variants of TGM either match or outperform GFN/PPO/SAC with a particularly pronounced difference in GFP, the largest domain. Valid. max refers to the object with the highest proxy reward in the validation set of the dataset used to train the proxy function.

As we have access to the ground-truth modes in this case, we evaluate methods according to the following metrics: (1) the number of modes found, where a mode is deemed found if a sample is generated within distance $\delta = 28$ of that mode; (2) for each mode, we track the distance to the closest sample generated during training. We report the average of this number over the $M$ modes for the best performing hyperparameters (see App. F.3 for further details). While the best GFN run matches q = 0, the increased peakiness of non-zero q-values yields noticeable benefits. The best-performing runs of q > 0 find samples that are much closer to the modes *while also discovering more of them*.

## 6.2 BIOLOGICAL SEQUENCE DESIGN [Q2]

We now focus on tasks based on real-world domains, where proxy rewards are trained on actual datasets. The tasks are based on the setup of Malkin et al. (2022) and use the datasets from Trabucco et al. (2022). For each, a transformer $\Phi$ is first trained on the training set (either for classification or regression). The transformer output is normalized before being used as a proxy reward (see App. F.4).

**5' Untranslated Region Sequence (UTR).** 5' UTR is an mRNA region that regulates transcription of the main coding sequence. The goal of the task is to find a UTR sequence of length 50 that maximizes predicted gene expression level. We take the dataset of 280 000 filtered sequences from Trabucco et al. (2022) and associated ribosome loads. We train a transformer as a regressor to predict the ribosome load from a sequence. The vocabulary consists of 4 nucleotides.

**Antimicrobial Peptide (AMP).** Antimicrobial peptides are short sequences of amino acids that have effects on microbes (bacteria, viruses, etc.). The goal is to discover novel peptides that are likely to have antimicrobial properties. The DBAASP database has collected known peptides with and without antimicrobial activity (Pirtskhalava et al., 2021). We use the dataset (sourced from Trabucco et al. (2022)) of 9222 non-AMP sequences and 6438 AMP sequences and train a binary classifier (predicting whether a sequence has antimicrobial properties or not). We use the normalized logit of the classifier as proxy reward. The vocabulary for this task consists of the 20 amino acids, plus a token corresponding to sequence termination. We mask the tokens such that the minimum sequence length is 14 and the maximum sequence length is 60. This is the only task where we allow variable-length sequences (as it is the only one whose dataset contains sequences of variable length).

**Green Fluorescent Protein (GFP).** The green fluorescent protein is a length-237 protein whose fluorescence has numerous applications in biology. The goal of this task is to discover other proteins with high predicted fluorescence. We take the processed dataset from Trabucco et al. (2022), which

Figure 5: Spread of final average mode rewards for various algorithms from a grid sweep over learning rates, $\beta$ and $\omega$. TGM on average performs better in AMP and GFP and similarly in UTR.

contains 56086 variations of the original sequence and their measured fluorescence. We train a regressor to predict the measured fluorescence, and the vocabulary consists of the 20 amino acids.

**Evaluation.** Each method is trained for $100\,000$ generated samples during which we regularly evaluate the learned network. By varying the temperature coefficient of the softmax representing the policy, we move along a quality/diversity curve for each sampler. To approximately evaluate Eq. 1, we generate a set of samples for a range of temperature coefficients. We then aggregate the samples and aim to approximately determine the best $k = 100$ distinct samples (i.e., such that the minimum distance is $\delta$) found by the policy. While this problem is NP-complete Karp (2009) (it can be seen as an instance of finding a maximal independent set), a greedy approach appears to work well in practice. For each method, a sweep is performed over hyperparameters. The best performing setting (based on final average mode reward) is run with 15 different seeds. See App. F.4 for further details.

**Results.** As illustrated in Tab. 2, for all three tasks, *all variants of TGM either roughly match or exceed the performance of GFN, soft actor-critic (SAC) (Haarnoja et al., 2018), and proximal policy optimization (PPO) (Schulman et al., 2017).* Similarly to what was found in (Malkin et al., 2022; Madan et al., 2023), SAC and PPO perform relatively poorly. We hypothesize that the credit assignment problem associated with terminal rewards is a significant issue for these methods. In AMP, where generating shorter sequences is possible, PPO is able to achieve similar results to TGM/GFN. The difference between TGM and GFN is most pronounced in the largest environment GFP, where TGM finds modes with significantly higher reward than the mean in Fig. 2. We hope this result shows the potential scalability of TGM. Interestingly, the effect of q is more varied in these environments, showing the benefits of interpolation in the GM operator. The best performing variants balance dilution and accumulation by using $q \in (0, 1)$.

### 6.3 HYPERPARAMETER ROBUSTNESS [Q3]

The best runs for Tab. 2 were selected from grid sweep over learning rates $\{0.00001, 0.0001, 0.001\}$, $\beta \in \{4, 16, 64, 256\}$ and inverse temperature parameters $\omega \in \{1, 4, 16\}$. For SAC and PPO, we set the entropy coefficient to $1/\omega$. For GFN, we set the sampling temperature of the policy to $1/\omega$. To ensure an equal number of runs per method, we fix $\alpha = 1$ and only vary $\omega$ for TGM. We expect that additionally varying $\alpha$ would further improve results.. We plot the performance distribution of this sweep in Fig. 5.

TGM variants seem relatively robust to these hyperparameter variations. The average SAC run is relatively stable but performs poorly. PPO is very sensitive to hyperparameter settings, with the best runs achieving strong performance in AMP. *The mean average mode reward of TGM is higher than GFN across* q *and environments*. Remarkably, TGM with q = 1 in AMP has little variability, with almost all runs performing well despite the significantly different hyperparameters. For GFP, there is significantly more variability, with the best performances coming from outliers for each method.

## 7 CONCLUSION

In this paper, motivated by the inadequacy of sampling proportional to reward, we generalize various soft RL operators and propose GM as an interpolated operator. From this operator, we develop a novel algorithm (TGM) and show it is consistently able to outperform GFNs on a variety of DCP tasks. Finally, we adapt the reward-robust RL framework to DCP tasks where rewards are given for entire trajectories to offer a new perspective on regularization in these problems. Ultimately, we believe TGM has the potential to be a more effective framework for finding promising candidates in scientific discovery applications. Testing TGM on language generation tasks with verifiable rewards and extending the method to general DAG structured DCPs remain exciting directions for future work.

## ACKNOWLEDGMENTS

The authors would like to thank Glen Berseth, Muqeeth Mohammed and Sobhan Mohammadpour for fruitful discussions and feedback as well as Moksh Jain for providing helpful advice for setting up the biological sequence/bit sequence experiments. This research was enabled in part by compute resources provided by Mila. G. Gidel is a CIFAR AI Chair, he is supported by a Discovery Grant from the Natural Science and Engineering Research Council (NSERC) of Canada. M. Jiralespong is supported by an IVADO fund under the Canada First Research Excellence Fund grant to develop robust, reasoning, and responsible artificial intelligence. E. Derman is supported by Samsung. M. Jiralerspong acknowledges the support of the Natural Sciences and Engineering Research Council of Canada (NSERC) [funding reference no. 600916].

## ETHICS STATEMENT

The purpose of this work is algorithmic in nature, and we do not aim to produce actual applicable models. Nonetheless, the ideas in this work could potentially be used to improve the training of models used in protein or other biological sequence design applications. While there are numerous beneficial applications for these types of models, they could also be used to design sequences that could eventually be harmful to humans. This harm could come from bad actors or even accidentally by over-optimizing to a single metric (e.g, antimicrobial resistance) and not considering potential harmful effects on the body. The extension of this work to multi-objective settings is a fruitful direction for future work.

In addition, the contribution of this work also has applications in language (another sequence design task). As such, it has the potential to improve the performance of large language models (LLMs) on both useful and nefarious tasks. Given the recent improvements in the abilities of these models, this potential effect is especially relevant.

## REPRODUCIBILITY STATEMENT

To promote transparency and reproducibility, we share our code implementation as well as checkpoints of the proxy reward functions we use. These allow for the reproduction of our results and fair future comparisons with new methods. We describe in ample detail the hyperparameters we use and design decisions we made in App. F as well as Sec. 6. See the supplementary material for the code/checkpoints.

## LLM USAGE STATEMENT

Claude Opus 4.6 was used for proofreading, spotting typos and improving wording/clarity for the camera-ready version of the paper.

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

APPENDIX

# A ROBUSTNESS-REGULARIZATION DUALITY

## A.1 CONVEX ANALYSIS

Before proving our results, we briefly recall notions of convex analysis below Bertsekas (2009).

**Definition A.1** (Convex conjugate). *Let a function $f : \mathbb{R}^n \to \bar{\mathbb{R}}$ with domain $\mathrm{dom}(f) \subseteq \mathbb{R}^n$. The convex conjugate of $f$ is defined as*

$$f^*(y) := \sup_{x \in \mathrm{dom}(f)} \langle x, y \rangle - f(x).$$

**Definition A.2** (Infimal convolution). *Given two functions $f, g : \mathbb{R}^n \to \bar{\mathbb{R}}$, the infimal convolution of $f$ and $g$ is defined as:*

$$[f \square g](x) := \inf_{z \in \mathbb{R}^n} \{ f(x - z) + g(z) \}.$$

**Property A.1** (Operations on conjugate transforms). *Let two functions $f_1, f_2 : \mathbb{R}^n \to \bar{\mathbb{R}}$ and a positive real number $\omega > 0$.*

   *(i) Defining $g(x) := \omega f_1(x)$, the convex conjugate of $g$ satisfies $g^*(y) = \omega f_1^*(\frac{y}{\omega})$.*

   *(ii) For $\kappa \neq 0$ and $h(x) := f_1(\kappa x)$, the convex conjugate of $h$ satisfies $h^*(y) = f_1^*(\frac{y}{\kappa})$.*

   *(iii) The convex conjugate of the sum $f_1 + f_2$ is the infimal convolution of their conjugates $f_1^* \square f_2^*$, namely:*

$$[f_1 + f_2]^*(y) = [f_1^* \square f_2^*](y), \qquad \forall y \in \mathbb{R}^n.$$

## A.2 APPLICATION TO POLICY DIVERGENCE

For Shannon entropy and KL divergence, the convex conjugate can be derived in closed form and is known to be a logsumexp function, so we omit the proof Geist et al. (2019); Derman et al. (2021).

**Proposition A.1** (Shannon conjugate). *Define the negative Shannon entropy $-\mathrm{H} : \mathbb{R}^n \to \bar{\mathbb{R}}$ as $[-\mathrm{H}](x) := \sum_{i=1}^n x_i \log(x_i)$ over the simplex domain $\Delta_n$. It is a convex function, and its convex conjugate is the logsumexp:*

$$\mathrm{LSE}_n(y) := \log \left( \sum_{i=1}^n e^{y_i} \right), \qquad \forall y \in \mathbb{R}^n.$$

**Proposition A.2** (KL conjugate). *Let $d \in \mathbb{R}^n$ be such that $d > 0$ and define the KL-divergence function $\mathrm{KL}_d(x) := \sum_{i=1}^n x_i \log \left( \frac{x_i}{d_i} \right), \quad \forall x \in \Delta_n$. It is a convex function, and its convex conjugate is the weighted logsumexp:*

$$\mathrm{WLSE}_{n,d}(y) := \log \left( \sum_{i=1}^n d_i e^{y_i} \right), \qquad \forall y \in \mathbb{R}^n.$$

We are now interested in deriving the Fenchel conjugate of a convex combination of Shannon entropy and KL divergence. Its explicit form is described below.

**Proposition A.3** (Shannon-KL conjugate). *For any $\mathsf{q} \in [0, 1]$, the convex combination $(1 - \mathsf{q})(-\mathrm{H}) + \mathsf{q}\mathrm{KL}_d$ satisfies:*

$$[(1 - \mathsf{q})(-\mathrm{H}) + \mathsf{q}\mathrm{KL}_d](x) = \sum_{i=1}^n x_i \log \left( \frac{x_i}{(d_i)^{\mathsf{q}}} \right), \qquad \forall x \in \mathbb{R}^n.$$

*and admits as convex conjugate the function:*

$$[(1 - \mathsf{q})(-\mathrm{H}) + \mathsf{q}\mathrm{KL}_d]^*(y) = \log \left( \sum_{i=1}^n (d_i)^{\mathsf{q}} e^{y_i} \right), \qquad \forall y \in \mathbb{R}^n.$$

*Proof.* The first statement comes from elementary algebra:

$$[(1-\mathsf{q})(-\mathrm{H}) + \mathsf{q}\mathrm{KL}_d](x) = (1-\mathsf{q})\sum_{i=1}^{n} x_i \log(x_i) + \mathsf{q}\sum_{i=1}^{n} x_i \log\left(\frac{x_i}{d_i}\right)$$

$$= (1-\mathsf{q}+\mathsf{q})\sum_{i=1}^{n} x_i \log(x_i) - \mathsf{q}\sum_{i=1}^{n} x_i \log(d_i)$$

$$= \sum_{i=1}^{n} x_i \log(x_i) - \sum_{i=1}^{n} x_i \log((d_i)^{\mathsf{q}})$$

$$= \sum_{i=1}^{n} x_i \log\left(\frac{x_i}{(d_i)^{\mathsf{q}}}\right) = \mathrm{KL}_{d^{\mathsf{q}}}(x).$$

For the second statement, we simply apply Prop. A.2 to obtain that

$$[(1-\mathsf{q})(-\mathrm{H}) + \mathsf{q}\mathrm{KL}_d]^*(y) = \mathrm{WLSE}_{n,d^{\mathsf{q}}}(y) = \log\left(\sum_{i=1}^{n}(d_i)^{\mathsf{q}}e^{y_i}\right).$$

A more involved proof would combine notions of infimal convolution with the respective conjugates of the two divergences. We omit it for brevity. $\square$

### A.3 PROOF OF THM. 5.1

We provide a slightly general proof below, with an arbitrary $\epsilon_s$-level set instead of just $0$.

**Theorem A.1** (Fenchel-Robust Regularized MDP). *For any $s \in \mathcal{S}$, define the reward set*

$$\mathcal{R}_s := r_0(s,\cdot) + \{r_s \in \mathbb{R}^{\mathcal{A}} : f_s^*(-r_s) \leq \epsilon_s\},$$

*and the regularization function $\Omega_s(\pi_s) := f_s(\pi_s) + \epsilon_s, \quad \forall s \in \mathcal{S}$. Then, for any policy $\pi \in \Pi$, $T_{\mathcal{R}}^{\pi}v = T^{\pi,\Omega}v, \quad \forall v \in \mathbb{R}^{\mathcal{S}}$, and $v_{\mathcal{R}}^{\pi} = v^{\pi,\Omega}$.*

*Proof.* First, we establish the support function of a set $\tilde{\mathcal{R}}_s := \{r_s' \in \mathbb{R}^{\mathcal{A}} : f_s^*(r_s') \leq \epsilon_s\}$ at any state $s \in \mathcal{S}$. By definition, for any policy $\pi_s \in \Delta_{\mathcal{A}}$, we have

$$\max_{r_s' \in \tilde{\mathcal{R}}_s} \langle \pi_s, r_s' \rangle = \max_{\{r_s' \in \mathbb{R}^{\mathcal{A}} : f_s^*(r_s') \leq \epsilon_s\}} \langle \pi_s, r_s' \rangle$$

$$= f_s(\pi_s) + \epsilon_s.$$

Next, we compute the robust Bellman operator associated with the uncertainty set $\mathcal{R} = \times_s \mathcal{R}_s$:

$$T_{\mathcal{R}}^{\pi}v(s) = \min_{r_s \in \mathcal{R}_s} r^{\pi}(s) + \gamma P^{\pi}v(s)$$

$$= \min_{r_s \in r_0(s,\cdot)+\tilde{\mathcal{R}}_s} r^{\pi}(s) + \gamma P^{\pi}v(s)$$

$$= r_0^{\pi}(s) + \min_{r_s' \in \tilde{\mathcal{R}}_s} \langle \pi_s, r_s' \rangle + \gamma P^{\pi}v(s)$$

$$= r_0^{\pi}(s) - \max_{r_s' \in \tilde{\mathcal{R}}_s} \langle \pi_s, -r_s' \rangle + \gamma P^{\pi}v(s)$$

$$= T_{r_0}^{\pi}v(s) - \max_{r_s' \in \tilde{\mathcal{R}}_s} \langle \pi_s, -r_s' \rangle.$$

It remains to compute

$$\max_{r_s' \in \tilde{\mathcal{R}}_s} \langle \pi_s, -r_s' \rangle = \max_{r_s'} \langle \pi_s, -r_s' \rangle \text{ s. t. } f_s^*(-r_s') \leq \epsilon_s$$

$$= \max_{\bar{r}_s} \langle \pi_s, \bar{r}_s \rangle \text{ s. t. } f_s^*(\bar{r}_s) \leq \epsilon_s$$

Employing the above expression of the support function enables us to write:

$$\max_{\bar{r}_s} \langle \pi_s, \bar{r}_s \rangle \text{ s. t. } f_s^*(\bar{r}_s) \leq \epsilon_s = f_s(\pi_s) + \epsilon_s$$

so that $T_{\mathcal{R}}^{\pi}v = T^{\pi,\Omega}v, \quad \forall v \in \mathbb{R}^{\mathcal{S}}$. The unique fixed point of each operator is $v_{\mathcal{R}}^{\pi}$ and $v^{\pi,\Omega}$, respectively, which leads to the conclusion. $\square$

Equation 13 comes from combining Thm. 5.1 with Prop. A.3. We can similarly obtain the uncertainty set resulting from KL and negative Shannon regularizations, based on their corresponding dual described in Sec. A.1. This leads us to the following table, summarizing the different regularizers used in this paper along with their corresponding uncertainty sets.

## A.4 PROOF OF PROP. 4.1

**Proposition A.4.** *For all* $q \in [0,1]$, $\Omega_s^q(\pi_s) = \frac{1}{\omega_s} \mathrm{KL}(\pi_s, d_s^{(q)}) - \frac{1}{\omega_s} \log(Z_q(d_s))$, *where* $Z_q(d_s) := \sum_{a \in \mathcal{A}} d_s(a)^q$ *and* $d_s^{(q)} = d_s^q / Z_q(d_s)$ *is the* $q$-*tilted softmax distribution.*

*Proof.* By definition,

$$
\begin{aligned}
\omega_s \Omega_s^q(\pi_s) &= [q\mathrm{KL}(\pi_s, d_s) + (1-q)(-\mathrm{H}(\pi_s))] \\
&= q \sum_{a \in \mathcal{A}} \pi_s(a) \log\left(\frac{\pi_s(a)}{d_s(a)}\right) + (1-q) \sum_{a \in \mathcal{A}} \pi_s(a) \log(\pi_s(a)) \\
&= -q \sum_{a \in \mathcal{A}} \pi_s(a) \log(d_s(a)) + \sum_{a \in \mathcal{A}} \pi_s(a) \log(\pi_s(a)) \\
&= \sum_{a \in \mathcal{A}} \pi_s(a) \log\left(\frac{\pi_s(a)}{d_s(a)^q}\right) \\
&= \sum_{a \in \mathcal{A}} \pi_s(a) \log\left(\frac{\pi_s(a)}{d_s(a)^q / Z_q(d_s)}\right) - \sum_{a \in \mathcal{A}} \pi_s(a) \log(Z_q(d_s)) \\
&= \sum_{a \in \mathcal{A}} \pi_s(a) \log\left(\frac{\pi_s(a)}{d_s(a)^q / Z_q(d_s)}\right) - \log(Z_q(d_s)),
\end{aligned}
$$

which yields the desired result. □

Table 3: Summary table of policy regularizers.

| | **Neg. Shannon** | **KL divergence** | **GSM** | **General convex** |
|---|---|---|---|---|
| **Regularizer** $\Omega_s$ | $-\mathrm{H}(\pi_s)$ | $\mathrm{KL}(\pi_s, d_s)$ | $\frac{1}{\omega_s}(q\mathrm{KL}(\pi_s, d_s) + (1-q)(-\mathrm{H}(\pi_s)))$ | $f_s(\pi_s) + \epsilon_s$ |
| **Conjugate** $\Omega_s^*$ | $\mathrm{LSE}_{\mathcal{A}}(q_s)$ | $\mathrm{WLSE}_{\mathcal{A}, d_s}(q_s)$ | $\omega_s^{-1}\mathrm{WLSE}_{\mathcal{A}, d_s^q}(\omega_s q_s)$ | $f_s^*(q_s) - \epsilon_s$ |
| **Reward Uncertainty** | $\{r_s : \Omega_s^*(-r_s) \leq 0\}$ | $\{r_s : \Omega_s^*(-r_s) \leq 0\}$ | $\{r_s : \Omega_s^*(-r_s) \leq 0\}$ | $\{r_s : f_s^*(-r_s) \leq \epsilon_s\}$ |

## A.5 COMPARISON WITH OTHER REGULARIZED RL WORKS

Eysenbach & Levine (2021) focuses on Shannon entropy, while we encompass a broad class of regularizers. Husain et al. (2021); Brekelmans et al. (2022) analyze robustness from the LP-dual perspective of RL. Although equivalent at optimum, this approach may not hold for any given policy. The notion of occupancy measure is also obscure in the context of DCP where transitions are fully determined by the agent's decisions and the time horizon is finite. Differently, Thm. 5.1 establishes

a robustness-regularization equivalence for any policy via Bellman evaluation operators. It thus provides a principled identity between robust dynamic programming in the sense of (Iyengar, 2005) and regularized MDPs in the sense of (Geist et al., 2019). Finally, the research motivation of Derman et al. (2021) being different, they proceed the opposite way from ours: they deduce a regularizer from generic uncertainty sets, whereas we deduce uncertainty sets from generic regularizers. This enables us to clarify the robustness properties caused by regularization.

# B  ADDITIONAL RESULTS

## B.1  IMPACT OF $\beta$ ON GFN PERFORMANCE

Another potential solution to the issue of sampling proportional to $e^{\Phi(x)}$ consists of modifying the reward function. In particular, the reward exponent hyperparameter $\beta$ could be used to arbitrarily increase the value of high-reward objects, hopefully overwhelming even an exponential amount of low-reward objects. To test this hypothesis, we perform an additional grid sweep for GFNs using higher values of $\beta$. We sweep over the same learning rates $\{0.00001, 0.0001, 0.001\}$ and sampling temperatures $\{1, 4, 16\}$ but also over $\beta \in \{512, 1024, 2048, 4096, 8192\}$. We plot the final average mode reward of the best performing setting for each $\beta$ in Fig. 6.

Going past 256 worsens performance for UTR and AMP but does noticeably improve performance for GFP. However, for all values of $\beta$, GFNs do not manage to reach the best-performing TGM setting. This discrepancy indicates TGM is exploring meaningfully different peaky distributions with better quality and diversity. Surprisingly, GFNs still perform decently even with very high $\beta$ values (up to 8192). It seems that gradient clipping and the VarGrad objective are enough to ensure some level of training stability at these high $\beta$ values. These values yield equally high losses, reaching values in the millions for $\beta = 8192$ on GFP.

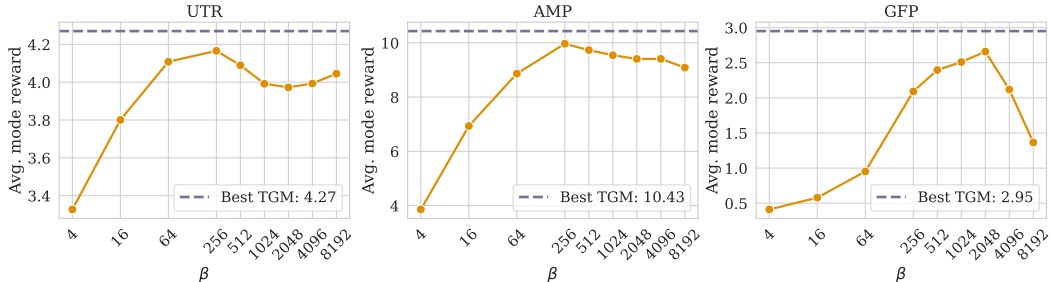

Figure 6: Average mode reward of best GFN setting for different values of $\beta$ compared to the best performing setting of TGM.

## B.2  COMPUTATIONAL COMPLEXITY

TGM is a generalization of TB but nearly identical in terms of computational complexity (essentially just requires an additional log-softmax in the loss computation). In particular, for sequence tasks with transformers, by acting on trajectories, TGM/TB are significantly faster than methods that act on transitions. The loss computation only requires a single forward pass per batch of trajectories instead of a forward pass per batch of transitions. We've included a table of training speed on an L40s GPU below. TGM/TB are roughly 10x faster than PPO/SAC when the latter splits generated trajectories into minibatches. TGM is only slightly more complicated to implement than GFNs, and simpler to implement than PPO (see the code submission for details).

Table 4: Wallclock training speed on an L40s for various methods on biological sequence tasks.

| Algorithm | UTR (samples/s) $\uparrow$ | AMP (samples/s) $\uparrow$ | GFP (samples/s) $\uparrow$ |
|---|---|---|---|
| SAC | 115 | 92 | 5 |
| PPO | 191 | 144 | 8 |
| TGM/TB | **1680** | **1216** | **120** |

## B.3  DIVERSITY AND NOVELTY EVALUATION

In addition to the aggregate metric of average mode reward, which seeks to approximate Eq. 1, we also evaluate the diversity and novelty of generated samples by each method. To do so, we use the metrics proposed in Jain et al. (2022). For diversity, the metric evaluates the mean pairwise

distance between generated samples. For novelty, the metric evaluates the mean minimum distance to a training set sample. We evaluate the modes generated by each method at the end of training and use the Levenshtein edit distance (Levenshtein et al., 1966).

Table 5: Diversity of TGM compared to baselines on the 3 biological sequence design tasks. Included is standard error over 15 seeds.

| | GFN | TGM (ours) | | | | |
|---|---|---|---|---|---|---|
| | | q = 0 | q = 0.25 | q = 0.5 | q = 0.75 | q = 1.0 |
| **UTR** | 18.38±0.06 | **18.75±0.12** | 18.29±0.12 | 18.58±0.09 | 18.62±0.07 | 18.20±0.11 |
| **AMP** | **17.76±1.18** | 16.54±0.83 | 14.82±0.60 | 13.69±0.05 | 14.06±0.75 | 12.94±0.03 |
| **GFP** | **149.90±2.42** | 84.18±2.63 | 89.95±7.59 | 106.68±7.69 | 96.36±6.58 | 107.83±5.46 |

Table 6: Novelty of TGM compared to baselines on the 3 biological sequence design tasks. Included is standard error over 15 seeds.

| | GFN | TGM (ours) | | | | |
|---|---|---|---|---|---|---|
| | | q = 0 | q = 0.25 | q = 0.5 | q = 0.75 | q = 1.0 |
| **UTR** | 31.99±0.12 | 30.97±0.06 | 30.85±0.06 | **32.37±0.10** | 32.09±0.07 | 30.94±0.07 |
| **AMP** | **36.21±4.14** | 35.87±4.05 | 27.65±3.33 | 21.11±0.02 | 24.65±2.38 | 20.94±0.03 |
| **GFP** | 205.65±0.26 | **212.48±0.78** | 210.38±0.98 | 209.86±0.67 | 211.03±0.63 | 210.93±0.60 |

In terms of diversity, GFNs and TGM obtain similar values for UTR. For AMP, diversity is higher for GFNs but this is partially due to all the modes GFN generates being of length 60. Since the Levenshtein edit distance is not normalized for distance, it is expected that the pairwise distance will grow with sequence length. Conversely, the sequence lengths of modes generated by TGM are more variable and shorter, limiting the potential edit distance. For GFP, GFNs obtain noticeably higher diversity, though at the cost of significantly lower reward.

For novelty, all methods are relatively similar for UTR and GFP. There is a significant difference for AMP, however this is likely due, once again, to the longer sequences being generated by GFNs. Overall, the increase in average mode reward does come at the cost of reduced diversity, though we argue this tradeoff is fair given the goals of scientific discovery (there is still enough diversity to have $k$ different interesting candidates).

### B.4 ADDITIONAL BASELINES

We compare the performance of TGM with 3 additional baselines: Munchausen DQN (Tiapkin et al., 2024; Vieillard et al., 2020), QGFN (Lau et al., 2024) and temperature-conditioned GFNs (Zhang et al., 2023; Kim et al., 2023). We note that QGFN and temperature-conditioned GFNs are complementary to our approach: the techniques they use can also be used with TGM.

Table 7: Mean mode reward of M-DQN, GFN and TGM as well as Q/temperature-conditioned variants. Included is the standard error over 5 seeds.

| | M-DQN | GFN | QGFN | GFN + Temp | Best TGM | QTGM | Best TGM + Temp |
|---|---|---|---|---|---|---|---|
| **UTR** | 3.37±0.12 | 4.12±0.00 | 4.08±0.04 | 4.18±0.05 | **4.27±0.01** | 4.05±0.06 | 4.18±0.01 |
| **AMP** | 2.97±0.82 | 10.04±0.04 | 8.96±0.44 | 9.97±0.12 | **10.43±0.07** | 9.41±0.25 | 9.83±0.34 |
| **GFP** | - | 1.90±0.03 | 1.08±0.07 | 1.89±0.04 | 2.95±0.08 | 0.99±0.14 | **3.30±0.15** |

### B.5 MUNCHAUSEN DQN

We set $\alpha = 0.1, l_0 = -20$ and separate the entropy coefficient from $\alpha$ as we observe that entropy coefficients $< 1$ tend to perform better (which would require $\alpha < 0$). Like other methods, we run a hyperparameter sweep: over learning rates $\in \{1e-5, 1e-4, 1e-3\}$, $\beta \in \{4, 16, 64, 256\}$ and entropy coefficients $\{1, 1/4, 1/16\}$ and select the best performing hyperparameters. We find that

Munchausen DQN (M-DQN) struggles to learn a good sampler. Similarly to SAC, the non-trajectory based loss appears to lead to unstable training and poor performance. A trajectory based version would likely address some of these issues.

## B.6 QGFN

We test the $p$-greedy QGFN setup (consisting of a mixture of forward policy with the argmax of the Q-values). We train a separate Q network in parallel (using the same architecture as the GFN) and set $n$ (in the $n$-step return) to be the maximum sequence length of the environment divided by 2. We sweep over learning rates $\in \{1e-5, 1e-4, 1e-3\}$, $\beta \in \{4, 16, 64, 256\}$ and values of $p \in \{0.25, 0.5, 0.75\}$ while fixing the sampling temperature to 1.

We also take the best performing hyperparameter setting from TGM and test it with the $p$-greedy setup (using $p = 0.25$). Doing so worsens performance in all 3 domains. While we have not tested them, it is possible that other variants of QGFN or the annealing schedule mentioned Lau et al. (2024) could improve performance. A more careful combination is beyond the scope of this work, but offers potentially interesting future avenues for research.

## B.7 Temperature-conditioned GFNs

We follow the method described in Kim et al. (2023) which expands on Zhang et al. (2023). Specifically, during training, for each batch we uniformly randomly sample a temperature between the set minimum temperature and maximum temperature. This temperature is encoded by a two-layer MLP before being added to each transformer layer. Then, as in Kim et al. (2023), the temperature is decoded into a scalar which is used as a temperature parameter, modulating the output logits. We then sweep over minimum temperatures $\{1/256, 1/64, 1/16, 1/4\}$ and maximum temperatures $\{1/128, 1/32, 1/8\}$ as well as learning rates in $\in \{1e-5, 1e-4, 1e-3\}$. For evaluation, we generate samples using the minimum temperature. Overall, this leads to similar or slightly better performance (on UTR) compared to a regular GFN.

We also take the best performing hyperparameter setting from TGM and test it with temperature conditioning between $\frac{1}{\beta}$ and $\frac{2}{\beta}$. The change results in a significant performance increase for GFP showing that temperature conditioning is a general technique that can be applied to both GFNs and TGM.

## C ACCUMULATION AND DILUTION

We prove the bounds in Tab. 1 below. For convenience, we denote $Q_s^* := \max_{a \in \mathrm{Ch}(s)} Q(s, a)$, $k := |\mathrm{Ch}(s)|$ and $\beta := \frac{q\alpha + \omega}{\alpha}$.

**Soft Bellman:** For the soft Bellman operator, $\Delta_s^{T^{\mathrm{SB}}} \geq 0$ since the logsumexp is always greater than or equal to the max. Then, since the logsumexp is strictly increasing, worse-case accumulation occurs when all the Q-values are equal, in which case $|\Delta_s^{T^{\mathrm{SB}}}| = \frac{1}{\omega} \log k e^{\omega Q_s^*} - Q_s^* = \frac{\log k}{\omega}$.

**Mellowmax:** For the mellowmax operator, $\Delta_s^{T^{\mathrm{MM}}} \leq 0$ since the logmeanexp is bounded by the max input. In the worst-case, we have the following bound $\frac{1}{\omega} \log \sum_{i=1}^{k} \frac{1}{k} e^{\omega Q_s(a_i)} \leq \frac{1}{\omega} \log \frac{e^{\omega Q_s^*}}{k} = Q_s^* - \frac{\log(k)}{\omega}$. Thus, $|\Delta_s^{T^{\mathrm{MM}}}| = -\Delta_s^{T^{\mathrm{MM}}} \leq \frac{\log(k)}{\omega}$.

### C.0.1 GENERAL MELLOWMAX

The general mellowmax operator $\Omega_\alpha^{q,*}$ can display accumulation or dilution depending on the value of $q$. To bound $|\Delta_s^{T^{\mathrm{GM}}}|$, we consider the cases $\Delta_s^{T^{\mathrm{GM}}}$ and $-\Delta_s^{T^{\mathrm{GM}}}$ separately. We show bounds for $\alpha > 0, \omega > 0, q \in (0, 1]$ and $k \in \mathbb{N}$ when $\beta > 1$.

We begin by noting that by factoring out $Q_s^*$ and denoting $\delta_i := Q(s, a_i) - Q_s^*$, we can rewrite $\Delta_s^{T^{\mathrm{GM}}}$ as follows. Without loss of generality, we assume the first Q-value is the largest, i.e., $Q_s^* = Q(s, a_1)$:

$$\Delta_s^{T^{\mathrm{GM}}} = \frac{1}{\omega} \log \langle \sigma(\alpha \mathbf{Q_s}))^q, e^{\omega Q_s} \rangle - Q_s^* = \frac{1}{\omega} \left[ \log(1 + \sum_{i=2}^{k} e^{(q\alpha + \omega)\delta_i}) - q \log(1 + \sum_{i=2}^{k} e^{\alpha \delta_i}) \right].$$
(14)

We can then make the change of variables $y_i = e^{\alpha \delta_i}$, which is continuous and bijective on $[0, +\infty)$. In particular, since $\delta_i \leq 0$, we are only interested in $y_i \in [0, 1]$. To simplify our analysis, we consider $g_\beta$ defined as follows

$$g_\beta(y_2, \ldots, y_k) := \log(1 + \sum_{i=2}^{k} y_i^\beta) - q \log(1 + \sum_{i=2}^{k} y_i),$$
(15)

with $\frac{1}{\omega} g_\beta(e^{\alpha \delta_2}, \ldots, e^{\alpha \delta_k}) = \Delta_s^{T^{\mathrm{GM}}}$.

**Accumulation bound:** We begin by showing that (15) is maximized at $y_2 = \ldots = y_k = 1$ for $y_i \in [0, 1]$. To see this, we consider the partial derivatives $\frac{\partial g_\beta}{\partial y_i} = \frac{\beta y_i^{\beta-1}}{1 + \sum_{i=2}^{k} y_i^\beta} - q \frac{1}{1 + \sum_{i=2}^{k} y_i}$ which is strictly positive for $y_i < 1$ (since $\beta > 1$). As a result, the function is increasing in all the $y_i$ on $[0, 1]$, implying the maximum is attained at the boundary. Hence,

$$\Delta_s^{T^{\mathrm{GM}}} \leq \frac{1}{\omega} g_\beta(1, \ldots, 1) = \frac{1}{\omega}(\log(k) - q \log(k)) = \frac{(1 - q) \log(k)}{\omega}.$$

**Dilution bound:** The dilution bound is more complex. Using the above, we have that $-\Delta_s^{T^{\mathrm{GM}}} = -\frac{1}{\omega} g_\beta(y_2, \ldots, y_k) \leq -\frac{1}{\omega} \min_{\{y_2, \ldots, y_k\} \in [0,1]^k} g_\beta(y_2, \ldots, y_k)$.

**Proposition C.1.** *Let $\alpha > 0$, $q \in (0, 1]$, $\omega > 0$ and $k \in \mathbb{N}$ with $\beta > 1$. The function*

$$g_\beta(y_2, \ldots, y_k) := \log(1 + \sum_{i=2}^{k} y_i^\beta) - q \log(1 + \sum_{i=2}^{k} y_i)$$

*attains a unique minimum on $[0, 1]^{k-1}$, and at this minimum, $0 < y_2 = \cdots = y_k < 1$.*

*Proof.* It is elementary to compute

$$\frac{\partial g_\beta}{\partial y_i} - \frac{\partial g_\beta}{\partial y_j} = \beta \left( \frac{y_i^{\beta-1} - y_j^{\beta-1}}{1 + \sum_{l=2}^{k} y_l^\beta} \right).$$

Since $\beta - 1 > 0$, for $(y_2, \ldots, y_k) \in [0, 1]^{k-1}$, $\frac{\partial g_\beta}{\partial y_i} > \frac{\partial g_\beta}{\partial y_j}$ if and only if $y_i > y_j$. This implies all local minima of $g_\beta$ in $[0, 1]^{k-1}$ are on the diagonal $y_2 = \cdots = y_k$.

To show the minimizer is unique, we now consider

$$h_\beta(y) := g_\beta(y, y, \ldots, y) = \log(1 + ny^\beta) - \mathsf{q} \log(1 + ny),$$

where $n := k - 1$. By computation, $h_\beta(0) = 0$ and $h_\beta(1) = (1 - \mathsf{q}) \log(1 + n) \geq 0$. To show the minimizer is in $y \in (0, 1)$ we need to show that $h_\beta(y) < 0$ for some $y \in (0, 1)$. Using $\frac{x}{1+x} \leq \log(1 + x) \leq x$ for $x > -1$, we get that

$$h_\beta(y) \leq ny^\beta - \frac{\mathsf{q}ny}{1 + ny} = ny \left( y^{\beta-1} - \frac{\mathsf{q}}{1 + ny} \right).$$

Taking $y = \frac{1}{n} \left( \frac{\mathsf{q}}{3} \right)^{\frac{1}{\beta-1}}$ we get

$$y^{\beta-1} - \frac{\mathsf{q}}{1 + ny} = \left( \frac{1}{n} \right)^{\beta-1} \frac{\mathsf{q}}{3} - \frac{\mathsf{q}}{1 + \left( \frac{\mathsf{q}}{3} \right)^{\frac{1}{\beta-1}}},$$

where the left term $\leq \frac{\mathsf{q}}{3}$ and the right term is $\geq \frac{\mathsf{q}}{2}$, satisfying $h_\beta(y) < 0$. Hence, since $h_\beta(0) = 0$, $h_\beta(y) < 0$ for some $y \in (0, 1)$ and $h_\beta(y) \geq 0$ for $y \geq 1$, by continuity, the minimum of $h_\beta$ on $[0, +\infty)$ exists, is attained at least once, and all minimizers are in $(0, 1)$. Furthermore, to show uniqueness, consider

$$h'_\beta(y) = \frac{\beta ny^{\beta-1}}{1 + ny^\beta} - \mathsf{q}\frac{n}{1 + ny} = \frac{n[ny^\beta(\beta - \mathsf{q}) + \beta y^{\beta-1} - \mathsf{q}]}{(1 + ny)(1 + ny^\beta)}. \tag{16}$$

We remark that the numerator of (16) is a strictly increasing function of $y$ for $y \geq 0$. Thus, $h'_\beta(y) = 0$ has at most one solution. $\qquad\square$

From Prop. C.1, we have that

$$- \min_{\{y_2, \ldots, y_k\} \in [0,1]^k} g_\beta(y_2, \ldots, y_k) = - \min_{y \in [0,1]} h_\beta(y) = \max_{y \in [0,1]} -h_\beta(y).$$

We aim to find an upper bound for $-h_\beta(y)$. For $c \geq 0$, $\beta \geq 1$, define

$$g(x, \beta, c) := -h_\beta(x) = \mathsf{q} \log(1 + cx) - \log(1 + cx^\beta).$$

**Proposition C.2.** *Let* $\mathsf{q} \in (0, 1]$, $\beta > 1$ *and* $c > 0$. *Let* $x^*(\beta, c)$ *and* $M(\beta, c)$ *be the argmax and max of* $g(\cdot, \beta, c)$ *(defined above) for* $x \in [0, 1]$, *respectively. Then, we have that*

$$M(\beta, c) \leq \mathsf{q} \left( 1 - \frac{1}{\beta} \right) \log(1 + c).$$

*Proof.* By Prop. C.1, $x^*(\beta, c)$ is well-defined and $0 < x^*(\beta, c) < 1$ for $\beta > 1$ and $c > 0$. $x^*(\beta, c)$ is not defined when $\beta = 1$ or $c = 0$ because $g(\cdot, 1, \cdot)$. Nonetheless, $M(\beta, c)$ is continuous from the right at $c = 0$, where it has limit 0, and at $\beta = 1$ where it has limit 0.

We denote by $g_x, g_\beta, g_c$ the partial derivatives of $g$. The constraint that $g_x(x^*(\beta), \beta, c) = 0$ simplifies to

$$\frac{\beta x^*(\beta, c)^{\beta-1}}{1 + cx^*(\beta, c)^\beta} = \frac{\mathsf{q}}{1 + cx^*(\beta, c)}. \tag{17}$$

Using this, we can simplify the derivative of $M$ with respect to $c$:

$$
\begin{aligned}
\frac{\partial M(\beta, c)}{\partial c} &= g_c(x^*(\beta, c), \beta, c) + \underbrace{g_x(x^*(\beta, c), \beta, c)}_{} \overbrace{\frac{\partial x^*(\beta, c)}{\partial c}}^{0} \\
&= \frac{\mathsf{q}x^*(\beta, c)}{1 + cx^*(\beta, c)} - \frac{x^*(\beta, c)^\beta}{1 + cx^*(\beta, c)^\beta} \\
&= \frac{\mathsf{q}x^*(\beta, c)}{1 + cx^*(\beta, c)}\left(1 - \frac{1}{\beta}\right) && \text{(Using (17))} \\
&\leq \frac{\mathsf{q}}{1 + c}\left(1 - \frac{1}{\beta}\right). && \text{(Using that } x^*(\beta, c) < 1)
\end{aligned}
$$

Integrating this inequality, for $c \geq 0$,

$$
M(\beta, c) \leq M(\beta, 0) + \int_0^c \frac{\mathsf{q}}{1 + t}\left(1 - \frac{1}{\beta}\right) dt = \mathsf{q}\left(1 - \frac{1}{\beta}\right)\log(1 + c).
$$

$\square$

Using Prop. C.2, we can plug in our parameters of interest by replacing $\beta$ with $\frac{\mathsf{q}\alpha + \omega}{\alpha}$ and $c$ with $k - 1$ to get

$$
M(\beta, c) \leq \mathsf{q}\left(\frac{\mathsf{q}\alpha + \omega - \alpha}{\mathsf{q}\alpha + \omega}\right)\log(k) \leq \mathsf{q}\left(\frac{\omega}{\mathsf{q}\alpha + \omega}\right)\log(k)
$$

Finally, getting back to $-\Delta_s^{T^{\mathrm{GM}}}$, we have

$$
-\Delta_s^{T^{\mathrm{GM}}} \leq -\frac{1}{\omega} \min_{\{y_2, \ldots, y_k\} \in [0,1]^k} g_\beta(y_2, \ldots, y_k) = \frac{1}{\omega} \max_{y \in [0,1]} -h_\beta(y) = \frac{M(\beta, k-1)}{\omega} \leq \frac{\log(k)}{\mathsf{q}\alpha + \omega}.
$$

Notably, this bound allows us to recover a worst-case bound on dilution of $\frac{\log(k)}{\alpha + \omega}$ for the soft mellowmax operator (where $\mathsf{q} = 1$). Combining bounds, we can solve for the $\mathsf{q}$ that yields the same bound for accumulation and dilution for $T^{\mathrm{GM}}$. Doing so yields $\mathsf{q} = \frac{\alpha - 2\omega + \sqrt{\alpha^2 + 4\omega^2}}{2\alpha}$.

## D    MULTI-STEP UNCERTAINTY SETS

While Fig. 3 examines the case of uncertainty sets for a single step, we now seek to explore the values that can be taken by $\delta_i$ throughout a trajectory and how this affects the final uncertainty set. For the entropic regularizer, if $\omega_s$ is the same for all $s$, then $\mathcal{R}_{s_i}$ is the same at every state. However, the final uncertainty set $\mathcal{R}$ for an object $x$ constructed through a trajectory of length $n$ is given by $\sum_i \mathcal{R}_{s_i}$ (i.e. the Minkowski sum of the individual uncertainty sets).

We extend the binary generation example to a DCP consisting of a sequence generation task with two tokens (i.e. a binary tree) and length $d$. Then, using Prop D.1, in Fig. 7, we illustrate how the sum of uncertainty sets changes as the depth of the tree increases for different operators.

**Proposition D.1.** *For a convex regularizer $f$ and associated robust set $\mathcal{R}_f = \{\delta \in \mathbb{R}^{|\mathcal{A}|} : f^*(\delta) \leq 0\}$, we have that:*

$$\sum_{i=1}^{k} \mathcal{R}_f = \{r \in \mathbb{R}^{|\mathcal{A}|} : kf^*\left(\frac{r}{k}\right) \leq 0\}. \tag{18}$$

*Proof.* We seek to show that, for a given $r \in \mathbb{R}^{|\mathcal{A}|}$, there exists $\delta_1, \delta_2 \ldots \delta_k$ such that $\sum_{i=1}^{k} \delta_i = r$ and $\delta_i \in \mathcal{R}_f$ if and only if $kf^*\left(\frac{r}{k}\right) \leq 0$.

[$\Rightarrow$]   Since $f$ is convex, so is $f^*$. Thus, using Jensen's inequality, we get that:

$$kf^*\left(\frac{r}{k}\right) = kf^*\left(\frac{\sum_i \delta_i}{k}\right) \leq \sum_i f^*(\delta_i) \leq 0,$$

where the last inequality stems from $\delta_i \in \mathcal{R}_f \implies f^*(\delta_i) \leq 0$.

[$\Leftarrow$]   Taking $\delta_i = \frac{r}{k}$ for all $\delta_i$, we have that $\sum_i \delta_i = r$. Then, $kf^*\left(\frac{r}{k}\right) \leq 0 \implies \forall i : f^*(\delta_i) = f^*\left(\frac{r}{k}\right) \leq 0$ and thus $\delta_i \in \mathcal{R}_f$. $\square$

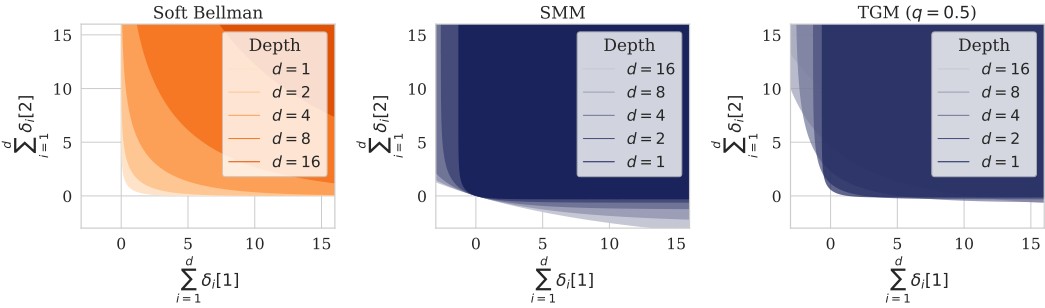

Figure 7: Illustration of sum of uncertainty sets along a trajectory. (Left) The final uncertainty set shifts farther away from $(0,0)$ as we increase $d$. As such, for objects that are deeper in the tree, the operator is only robust to objects having much higher reward than their proxy evaluation. This behavior provides an alternative viewpoint for the preference of the soft Bellman operator for longer objects. (Middle) On the other hand, the SMM operator stays centered around $(0,0)$ and becomes increasingly robust to decreases in rewards. (Right) The final uncertainty set of the GM operator stays close to $(0,0)$ and is robust to some decreases in reward.

# E TRAJECTORY GENERAL MELLOWMAX

## E.1 TGM HYPERPARAMETERS

We summarize the TGM hyperparameters (with reward being given by $e^{\beta\Phi(x)}$) below. Recall that TGM solves the following regularized problem:

$$\arg\max_{\pi_s}\langle\pi_s, Q_s\rangle - \frac{1}{\omega}\left[\mathsf{q}\mathrm{KL}(\pi_s, \mathrm{softmax}(\alpha\mathbf{Q_s}))) + (1-\mathsf{q})(-\mathrm{H}(\pi_s))\right],$$

whose corresponding operator is given by

$$\Omega_\alpha^{\mathsf{q},*}(\mathbf{Q_s}) = \frac{1}{\omega}\log\langle\mathrm{softmax}(\alpha\mathbf{Q_s}))^{\mathsf{q}}, e^{\omega Q_s}\rangle.$$

| Parameter | Interpretation | Range |
|---|---|---|
| $\alpha$ | Temperature of the softmax of the Q-values used by KL regularizer | $\mathbb{R}$ |
| $\omega$ | Regularization coefficient | $\mathbb{R}$ |
| $\mathsf{q}$ | Interpolation coefficient between entropy and KL regularization | $[0,1]$ |
| $\beta$ | Exponent of the reward function | $\mathbb{R}$ |

Table 8: Summary of TGM hyperparameters and their interpretation.

## E.2 PROOF OF OPTIMALITY

**Notation:** We denote by $\mathrm{Ch}(s)$ the set of possible actions from state $s$. $\sigma_\tau(Q_s) := \frac{e^{\tau Q_s}}{\sum_{a'\in\mathrm{Ch}(s)}e^{\tau Q_s[a']}}$ is the softmax with inverse temperature $\tau$. For a DCP and a given state $s$, we denote by $s'_a$ the state reached by taking action $a$ (unique since transitions are deterministic). We denote by $E$ the set of edges in $\mathcal{G}$, each of which represents a transition $(s, a, s'_a)$. Finally, for convenience, we denote

$$g(Q_s, a) := \frac{1}{\omega}\log\sigma_{\mathsf{q}\alpha+\omega}(Q_s)[a] - \mathsf{q}\log\sigma_\alpha(Q_s)[a],$$

$$g^*(Q_s) := \frac{1}{\omega}\log\langle\sigma_\alpha(Q_s)^{\mathsf{q}}, e^{\omega Q_s}\rangle.$$

**Proposition E.1.** *In a DCP $(\mathcal{G}, r)$, for any state $s$,*

$$v_s^* := \max_{\pi_s}\langle\pi_s, Q_s\rangle - \tfrac{1}{\omega}\Omega_\alpha^{\mathsf{q}}(\pi_s) = \frac{1}{\omega}\log\langle\sigma_\alpha(Q_s)^{\mathsf{q}}, e^{\omega Q_s}\rangle = g^*(Q_s), \tag{19}$$

*and*

$$\pi_s^* = \sigma_{\mathsf{q}\alpha+\omega}(Q_s). \tag{20}$$

*is the unique maximizer of the state-wise regularized problem.*

*Proof.* Unlike the case of a generic $\pi$, (19) is not a convex conjugate since $Q_s$ appears in both the dot product and $\Omega$. Nonetheless, $\Omega$ is still convex in $\pi$. Thus, similarly to (Nachum et al., 2017), we consider the Lagrangian:

$$L(\pi_s, \lambda) = \langle\pi_s, Q_s\rangle - \tfrac{1}{\omega}\Omega_\alpha^{\mathsf{q}}(\pi_s) + \lambda(1 - \langle\mathbf{1}, \pi_s\rangle).$$

Since $\Omega$ is convex, the overall function to optimize is concave. Given that Slater's condition holds, the KKT conditions are both sufficient and necessary for optimality. Due to the lack of cycle in DCPs, we have that $\nabla_{\pi_s}Q_s = 0$. Hence, the KKT conditions yield the following system of equations:

$$0 = Q_s - \frac{1}{\omega}\left(\mathsf{q}[\log\pi_s - \log\pi_{Q_s} + 1] + (1-\mathsf{q})[\log\pi_s + 1]\right) - \lambda \tag{21}$$

$$1 = \langle\pi_s, \mathbf{1}\rangle \tag{22}$$

where we denote $\pi_{Q_s} := \sigma_\alpha(Q_s)$. Solving for $\pi_s$ in the first, we get that the optimal policy $\pi_s^*$ is given by:

$$\pi_s^* = \exp([Q_s - \lambda]\omega + \mathsf{q}\log\pi_{Q_s} - 1)$$

which ensures that $\pi_s \geq 0$. Replacing the above in (22), we have that:

$$\lambda^* = \frac{1}{\omega}\left(\log\sum_{a\in\text{Ch}(s)} e^{(\alpha\mathsf{q}+\omega)Q_s[a]} - \mathsf{q}\log\sum_{a\in\text{Ch}(s)} e^{\alpha Q_s[a]}\right) = \frac{1}{\omega}\left(\log\langle\sigma_\alpha(Q_s)^\mathsf{q}, e^{\omega_s Q_s}\rangle - 1\right).$$

We note the similarity of $\lambda^*$ and the maximal value (19). Plugging $\lambda^*$ back in to $\pi_s^*$, we get

$$\pi_s^* = \sigma_{\mathsf{q}\alpha+\omega}(Q_s). \tag{20}$$

Finally, plugging $\pi_s^*$ in $\langle\pi_s, Q_s\rangle - \frac{1}{\omega}\Omega_\alpha^\mathsf{q}(\pi_s)$ with some straightforward algebraic manipulations, we obtain (19). Since both the KL divergence and the negative entropy are strictly convex, (20) is the unique maximizer. $\quad\square$

**Corollary E.1.** *For any DCP $(\mathcal{G}, \Phi)$, the GM operator has unique optimal value/policy functions $(v^*, \pi^*)$.*

*Proof.* We show this by induction on an inverse topological sorting $s_0, s_1 \ldots, s_n$ of the states of $\mathcal{G}$. For a state with no children, the optimal value is defined to be 0 and there is no policy. Then, assume each state $s_i, i < N$ has a unique optimal value/policy. By the properties of the topological sorting, all the children of $s_N$ have a unique optimal value/policy. Then, by Prop. E.1, $(v_{s_N}^*, \pi_{s_N}^*)$ is also uniquely defined. $\quad\square$

**Lemma E.1.** *For any $Q_s$ and $a$, we have that*

$$g(Q_S, a) = Q_s[a] - g^*(Q_s). \tag{23}$$

*Proof.* By simple algebra,

$$
\begin{aligned}
&g(Q_s, a)\\
&= \frac{1}{\omega}(\log\sigma_{\mathsf{q}\alpha+\omega}(Q_s)[a] - \mathsf{q}\log\sigma_\alpha(Q_s)[a])\\
&= \frac{1}{\omega}\left((\mathsf{q}\alpha+\omega)Q_s[a] - \mathsf{q}\alpha Q_s[a]\right)\\
&\quad -\frac{1}{\omega}\left(\log\sum_{a'\in\text{Ch}(s)} e^{(\mathsf{q}\alpha+\omega)Q_s[a']} - \log\left[\sum_{a'\in\text{Ch}(s)} e^{\alpha Q_s[a']}\right]^\mathsf{q}\right)\\
&= \frac{1}{\omega}(\omega Q_s[a]) - \frac{1}{\omega}\log\left[\sum_{a'\in\text{Ch}(s)}\frac{e^{\mathsf{q}\alpha Q_s[a']}e^{\omega Q_s[a']}}{\left(\sum_{a'\in\text{Ch}(s)} e^{\alpha Q_s[a']}\right)^\mathsf{q}}\right]\\
&= Q_s[a] - \frac{1}{\omega}\left(\log\langle\sigma_\alpha(Q_s)^\mathsf{q}, e^{\omega_s Q_s}\rangle\right)\\
&= Q_s[a] - g^*(Q_s)
\end{aligned}
$$

$\quad\square$

**Lemma E.2.** *For any state $s$ in a DCP, the following are equivalent:*

$$\sigma_{\alpha\mathsf{q}+\omega}(Q_s^\theta) = \arg\max_{\pi_s}\langle\pi_s, Q_s^*\rangle - \frac{1}{\omega}\Omega_{\sigma_\alpha(Q_s^*), s}^\mathsf{q}(\pi_s), \tag{24}$$

$$\forall a \in \text{Ch}(s): v_{s_a'}^* = v_s^* + g(Q_s^\theta, a) - r(s, a). \tag{25}$$

*Proof.*

**[ (24) $\Rightarrow$ (25)]** Since $\pi_s^*$ is unique, by (24) and (20), by the properties of the softmax, we have that $\sigma_{\alpha q + \omega}(Q_s^\theta) = \sigma_{q\alpha+\omega}(Q_s^*)$ and $\sigma_\alpha(Q_s^\theta) = \sigma_\alpha(Q_s^*)$. Then, for any $a \in \mathrm{Ch}(s)$:

$$
\begin{aligned}
& v_s^* + g(Q_s^\theta, a) \\
&= v_s^* + \tfrac{1}{\omega} \left( \log \sigma_{q\alpha+\omega}(Q_s^\theta)[a] - q \log \sigma_\alpha(Q_s^\theta)[a] \right) \\
&= \tfrac{1}{\omega} \left( \log \langle \sigma_\alpha(Q_s^*)^q, e^{\omega_s Q_s^*} \rangle + \log \sigma_{q\alpha+\omega}(Q_s^*)[a] - q \log \sigma_\alpha(Q_s^*)[a] \right) && \text{Using (19).} \\
&= \tfrac{1}{\omega} \left( \log \frac{\sum_{a' \in \mathrm{Ch}(s)} e^{(q\alpha+\omega)Q_s^*[a']}}{\sum_{a' \in \mathrm{Ch}(s)} e^{\alpha Q_s^*[a']}} + \log \sigma_{q\alpha+\omega}(Q_s^*)[a] - q \log \sigma_\alpha(Q_s^*)[a] \right) \\
&= \frac{1}{\omega} \left( \log e^{(q\alpha+\omega)Q_s^*[a]} - q \log e^{\alpha Q_s^*[a]} \right) \\
&= Q_s^*[a] \\
&= v_{s_a'}^* + r(s, a) && \text{Using the definition of } Q_s[a].
\end{aligned}
$$

**[ (24) $\Leftarrow$ (25)]** For any $a \in \mathrm{Ch}(s)$, we have that:

$$
\begin{aligned}
v_{s_a'}^* &= v_s^* + g(Q_s^\theta, a) - r(s, a) \\
\Longleftrightarrow \quad v_{s_a'}^* &= v_s^* + Q_s^\theta[a] - g^*(Q_s^\theta) - r(s, a) && \text{Using (23)} \\
\Longleftrightarrow \quad v_s^* - g^*(Q_s^\theta) &= Q_s^*[a] - Q_s^\theta[a]
\end{aligned}
$$

Since the above equality holds for all actions and $v_s^*$ and $g^*(Q_s^\theta)$ do not depend on $a$, we have that $Q_s^*[a] - Q_s^\theta[a]$ must be equal for all actions. Thus, $\forall a \in \mathrm{Ch}(s) : Q_s^*[a] - Q_s^\theta[a] = c$ which implies

$$
Q_s^\theta[a] = Q_s^*[a] - c, \tag{26}
$$

for some action independent value $c$. Finally, we have

$$
\sigma_{\alpha q + \omega}(Q_s^\theta) = \sigma_{\alpha q + \omega}(Q_s^* - c) = \sigma_{\alpha q + \omega}(Q_s^*),
$$

by the properties of the softmax. $\qquad\square$

### E.3 PROOF OF THEOREM 4.1

We note that it is straightforward to derive the equivalent subtrajectory constraint.

**Theorem E.1** (Trajectory GM). *For any DCP $(\mathcal{G}, \Phi)$, let $(v^*, \pi^*)$ be the unique optimal value/policy functions for the GM operator. Then, for a given Q-function $Q^\theta$,*

$$
\sigma_{\alpha q + \omega}(Q_s^\theta) = \arg\max_{\pi_s} \langle \pi_s, Q_s \rangle - \tfrac{1}{\omega} \Omega_\alpha^q(\pi_s), \tag{9}
$$

*holds for all states $s$ if and only if*

$$
v_0^* + \sum_{i=0}^n \underbrace{\frac{1}{\omega} \left( \log \sigma_{q\alpha+\omega}(Q_{s_i}^\theta)[a_i] - q \log \sigma_\alpha(Q_{s_i}^\theta)[a_i] \right)}_{g(Q_{s_i}^\theta, a_i)} - r(s_i, a_i) = 0, \tag{10}
$$

*holds for all full trajectories $s_0 \xrightarrow{a_0} \cdots \xrightarrow{a_{n-1}} s_n \xrightarrow{a_n} x$ in the DCP.*

*Proof.* Similarly to (Nachum et al., 2017), we aim to show that optimality implies consistency and vice versa. A refresher of relevant notation can be found at the start of App. E.

**Optimality implies consistency** [ (9) $\Rightarrow$ (10)] From Lemma E.2, for any given trajectory, we have that (25) holds for all transitions in the trajectory. Then, (10) follows straightforwardly by expanding (25) over that trajectory in a recursive manner.

**Consistency implies optimality**    [ (9) $\Leftarrow$ (10)] Consider an inverse topological sorting $s^1, s^2 \ldots s^M$ of $\mathcal{G}$ such that for any state $s^i$, any child $s^j$ of $s^i$ has index $j < i$.

Now, we seek to show by induction over this sorting that the following holds for any state $s$ and subtrajectory $s_0 \to s_1 \ldots \to s_n \to s$ ending in $s$:

$$v_0^* + \sum_{i=0}^{n} \frac{1}{\omega} \left( \log \sigma_{\mathsf{q}\alpha+\omega}(Q_{s_i}^\theta)[a_i] - \mathsf{q} \log \sigma_\alpha(Q_{s_i}^\theta)[a_i] \right) - r(s_i, a_i) = v_s^*. \tag{27}$$

For the base case $s^1$, $s^1$ must be a leaf and belong to $\mathcal{X}$. Then, by assumption $v_{s^1} = 0$ and (27) holds by (10). Now, suppose that (27) holds for any $s^i$, $i < N$. Then, by the definition of the sorting, for all $a \in \mathrm{Ch}(s^N)$, (27) holds for $s_a$ where $s^N \xrightarrow{a} s_a$.

Let $s_0 \to s_1 \ldots \to s_n \to s^N$ be a trajectory ending in $s^N$. From the induction hypothesis, $\forall a \in \mathrm{Ch}(s)$ :

$$v_0^* + \sum_{i=0}^{n} g(Q_{s_i}^\theta, a_i) - r(s_i, a_i) + g(Q_{s^N}^\theta, a) - r(s^N, a) = v_{s_a}^*$$

$$v_0^* + \sum_{i=0}^{n} g(Q_{s_i}^\theta, a_i) - r(s_i, a_i) - g^*(Q_{s^N}^\theta) = Q_{s^N}^*[a] - Q_{s^N}^\theta[a] \quad \text{Using (23).}$$

Similarly to the proof of Lemma E.1, we have that the LHS is independent of $a$ but must be equal to $Q_{s^N}^*[a] - Q_{s^N}^\theta[a]$ for all $a$. As such, we are guaranteed

$$Q_{s^N}^\theta[a] = Q_{s^N}^*[a] - c$$

for some action independent value $c$ and all $a \in \mathrm{Ch}(s^N)$. Then, it is straightforward to compute $g^*(Q_{s^N}^\theta) = g^*(Q_{s^N}^* - c) = g^*(Q_{s^N}^*) - c$. Plugging both back in and canceling $c$, we get

$$v_0^* + \sum_{i=0}^{n} g(Q_{s_i}^\theta, a_i) - r(s_i, a_i) = g^*(Q_{s^N}^*) = v_{s^N}^*$$

and the induction is complete. Finally, it remains to show that $\sigma_{\alpha\mathsf{q}+\omega}(Q_s^\theta)$ is optimal for all $s$. If $s$ has no children, optimality holds trivially. If $s$ has at least one child $a$, take a trajectory $s_0 \to s_1 \ldots \to s_n \to s$ ending in $s$. From (27), we have that both:

$$v_0^* + \sum_{i=0}^{n} g(Q_{s_i}^\theta, a_i) - r(s_i, a_i) = v_s^*$$

$$\forall a \in \mathrm{Ch}(s) : v_0^* + \sum_{i=0}^{n} g(Q_{s_i}^\theta, a_i) - r(s_i, a_i) + g(Q_s^\theta, a) - r(s, a) = v_{s_a}^*.$$

By subtracting the first from the second, we get that, $\forall a \in \mathrm{Ch}(s)$:

$$g(Q_s^\theta, a) - r(s, a) = v_{s_a}^* - v_s^*$$

$$v_s^* + g(Q_s^\theta, a) - r(s, a) = v_{s_a}^*.$$

By Lemma E.2, $\sigma_{\alpha\mathsf{q}+\omega}(Q_s^\theta)$ is thus optimal at $s$ and we are done. $\qquad \square$

# F  EXPERIMENTAL DETAILS

## F.1  ALGORITHMS

We follow most of the hyperparameter decisions of (Malkin et al., 2022). For each method, we train a transformer with 3 layers, 8 attention heads, and an embedding dimension of 64. We set dropout to 0.1 and use causal masking. Each output head uses a fully-connected network with 2 hidden layers of dimensions 256.

We use weight decay of $10^{-4}$ and use gradient clipping with a threshold of 10 to help stabilize training at higher $\beta$ values. When generating samples for training, we use a policy corresponding to a mixture of the method's policy and a uniform policy (with weight 0.01). Each method is trained with Adam (Kingma, 2014) with $\epsilon = 10^{-5}$.

**TGM/GFN:**  Since GFNs correspond to a specific hyperparameter setting for TGM, training details are essentially identical between the two methods. Both methods are trained online with a batch size of 16 trajectories and have a single head that outputs logits. We sample trajectories by taking the tempered softmax of the logits, using the temperature $\alpha q + \omega$ from the optimal policy  (20). For GFNs, $\omega$ affects the sampling policy but *not the actual training objective* as this ensures the network is learning the actual GFN objective.

**SAC:**  For SAC, we use three networks: two Q networks (to reduce overestimation bias) and a policy network, following the implementation of (Christodoulou, 2019; Huang et al., 2022). We use a replay buffer of size $100\,000$ and a batch size of $1\,024$ transitions. Generation/training is adjusted to have a replay ratio of 1.0. The entropy coefficient is fixed at $1/\omega$, and we use target networks (for the Q networks), which get updated every 10 iterations.

**PPO:**  For PPO, we use two networks: a policy network and a value network, following the implementation of (Huang et al., 2022). We follow the hyperparameter settings of (Huang et al., 2022) and use a generalized advantage estimator $\lambda$ of 0.95 and $\gamma = 1$ to avoid discounting. The clipping epsilon is 0.1 and the value coefficient is 0.5. During online training, 16 trajectories are generated, and the resulting transitions are split into 4 minibatches for training. The entropy coefficient is set to $1/\omega$.

## F.2  SEQUENCES

For each sequence generation task, we add the following 3 tokens to the vocabulary: BOS, PAD, EOS. Sequences are padded to be of length equal to the maximum length for the task, with an added BOS, EOS. Sequences look like the following: $[\text{BOS}, x_1, \ldots, x_n, \text{PAD}, \ldots, \text{PAD}, \text{EOS}]$. When generating sequences, they start with BOS, and tokens are added until a terminating action is chosen, at which point PAD/EOS are added automatically. The terminating action is masked until the task's minimum length is reached.

## F.3  SYNTHETIC TASKS

**TF-Bind-8.**  Hyperparameters are as described in the paper. We sample $10\,000$ sequences for each method using their respective optimal policy. Fig. 4 is the kernel density estimate computed using these samples with the default parameters used by seaborn (Waskom, 2021).

**Bit sequence.**  We use $n = 120$ and $k = 8$ with $M = 60$ modes and force sequences to be of length 120 through action masking. The modes are sampled randomly and fixed across runs to ensure there is no variation from certain runs having harder-to-find modes. Each method is trained for $200\,000$ samples.

## F.4  BIOLOGICAL SEQUENCE DESIGN

For each task, once the proxy reward model is trained, we compute the mean and standard deviation of the logits on the validation set. Then, we use the following normalized value as reward

$$r(x) = \beta\Phi(x) = \beta\frac{\phi(x) - \mu_{\text{val}}}{\sigma_{\text{val}}}, \tag{28}$$

where $\phi(x)$ is the logit output by the trained model. To train the proxy reward models, we follow the hyperparameters used by (Malkin et al., 2022).

**UTR:** Using the described dataset, we train a transformer with 4 layers, 8 attention heads, and an embedding dimension of 64. 80% of the data is used for training and the rest for validation. We train for 250 epochs with early stopping (using a patience of 15). The learning rate is set to $10^{-4}$, and we use a batch size of 128 and weight decay of 1e-6. We set both the minimum and maximum length to 50 since all sequences in the dataset are of length 50.

**AMP:** We use the same hyperparameters for training the model as UTR. Since the dataset has sequences of variable length, we set the minimum length to 14 and the maximum length to 60. Unlike the other tasks, the model is trained for binary classification, and we use the logit passed to the sigmoid as $\phi(x)$.

**GFP:** GFP is the largest task. We train a transformer with 3 layers, 8 attention heads, and an embedding dimension of 128. Since all sequences are of length 237, we set both the minimum length and maximum length to 237.

### F.4.1 EVALUATION

For evaluation of the average mode reward, we sweep over inverse temperature modifiers $t \in \{0.005, 0.01, 0.02, 0.05, 0.1, 0.2, 0.5, 1, 2, 5\}$ (i.e. if the regular sampling policy was $\pi \propto e^{\tau x}$, we sample using $\pi' \propto e^{\tau t x}$) and generate 512 samples for each. We then aggregate the samples and greedily find the top 100 samples such that they are at least at a distance $d_{\min}$ of each other. We use the Levenshtein edit distance as metric (Levenshtein et al., 1966) and set $d_{\min} = 0.25 \cdot \frac{\text{minLen}+\text{maxLen}}{2}$ (i.e. $d_{\min} = 13$ for UTR, $d_{\min} 10$ for AMP and $d_{\min} = 60$ for GFP). The constraint can be roughly interpreted as requiring at least 25% of a sequence to be different for it to be considered distinct. In Tab. 2, we report the max of this value achieved over the course of training.

### F.5 COMPUTE RESOURCES

Experiments were conducted on a cluster with a mix of GPUs. All experiments were run using jobs that requested 1xL40s, 24G of RAM, and 4 CPUs. The cluster allows for dozens of jobs to be run in parallel. Overall, the experiments (including all the hyperparameter sweeps, baselines, and various seeds) used roughly:

- **Bit Sequence:** 35 GPU hours.
- **UTR:** 48 GPU hours.
- **AMP:** 48 GPU hours.
- **GFP:** 300 GPU hours.

for a total of roughly $\approx 20$ GPU days. In particular, the entire codebase uses Jax (Bradbury et al., 2018) and a parallelized environment allowing for fast generation. Since we use transformers, training on a generated trajectory for TGM/GFNs only requires a single forward/backward pass. This significantly speeds up training (roughly 4x faster) compared to the PPO/SAC baselines, which operate on individual transitions (though there are potential engineering optimizations to improve the speed of these baselines).

