# OpenReview forum: "Discrete Compositional Generation via General Soft Operators and Robust Reinforcement Learning"
_ICLR.cc/2026/Conference — ICLR 2026 Poster_

### Official Review · Reviewer_iNss · 2025-10-31

**Soundness:** 3
**Presentation:** 3
**Contribution:** 3
**Rating:** 6
**Confidence:** 2

**Summary:**

The paper addresses the problem of constructing/composing discrete objects that maximize an uncertain (learnt) proxy reward and are diverse. The starting point of the paper is Generative Flow Networks (GFNs) which learn to sample proportionally to some reward function. However the number of possible discrete objects is exponential to their length/number of their components. Thus sampling proportionally to the value of the reward function will tend to produce samples that are of low reward, simply because low reward samples will have collectively a larger probability of being selected than the few high reward samples. Thus the problem that the paper sets to address is given equation 1 of the paper:
$$ \max_{x_1, ..., x_k} \sum_i \exp{\Phi(x_i)} \text{ subject to } d(x_i, x_j) > \delta, \text{ eq. 1 } $$
where $\Phi(.)$ is the learnt reward function and $d(.,.)$ some distance measure meant to capture diversity in the set of selected structures.

The paper casts this problem as learning a policy that is robust to the worst-case reward model, i.e. the value function that they seek to maximize is:
$$v_{\mathcal R}^{\pi} (s) := \min_{ r \in \mathcal R} v^{\pi}(s)$$
Uncertain rewards create reward uncertainty sets and a robust policy should be optimal under the worst case reward; however at least as eq. 1 above is stated the initial optimization problem that the paper sets to address looks rather like a standard optimization problem and not a robust optimization problem.

Given that one should learn a robust policy to address 1 then the paper proceeds with the following contributions:
* a regulariser that explicitly trades off between different regularisation functions associated with GFN, and ((soft)mellowmax),
* an algorithm that uses the above regulariser and operates on the level of trajectories and samples from high reward regions
* a characterisation of the reward uncertainty sets (created by the uncertain reward) that are induced by convex regularises

The paper discusses the relation between GFNs, soft Bellman, and the (soft)-mellow max operators:
* There is a direct equivalence between GFNs and soft Bellman (entropy regularised MDP/policy $H(\pi)$, both suffer from value accumulation: the value accumulation of a large number of suboptimal objects can overwhelm a single high reward object
* The mellowmax operator solves accumulation but suffers from dilution; a high reward object can get diluted by lower reward objects.
* The soft mellowmax addresses the dilution of mellowmax

The paper argues for controlling the balance of accumulation and dilution and does so by addressing a regulariser that allows to seamslessly transition between the three operators.  GFNs and soft Bellman are entropy regularised MDP/policy $H(\pi)$); mellowmax/soft-mellowmax learn  KL regularised policies, $KL(\pi,d_s)$, where $d_s$ are the uniform distribution and the $softmax(Q)$ distributions respectively. The paper introduces the general mellow max regulariser that interpolates between the two as:
$$ q KL(\pi,d_s) + (1-q) (-H(\pi_s)) \text{ eq. 6 } $$
interpolating between policy entropy maximization and getting a policy that is close to the softmax of the $Q$ values, capturing GFNs, and (soft)-mellowmax for different values of $q$ and temperature.

The paper then proceeds to introduce the trajectory general mellow max which builds upon eq 6 and operates over trajectories. The algorithm is evaluated over a set of synthetic and real world sequence generation benchmarks that show that TGM achieves high rewards against RL baselines, namely PPO and SAC, and it is on a par or better than GFN. The experiments show that TGM has a performance that is quite stable with respect to difference values of the interpolation coefficient .



In addition the paper discusses the equivalence of entropy regularised MDPs to robust MDPs with uncertain reward (robust RL).

The paper proceeds with a discussion of the uncertainty sets introduced by the different regularisers, i.e. the entropy regulariser used in GFNs, the softmellowmax, and the general mellowmax. In particular the setting of discovering interesting structures one would like to maximize the proxy reward $\Phi$ while accounting for the difference, $\delta$, between $\Phi$ and the true uknown reward $r^*$, by learning a sampling distribution $p$ that satisfies:
$$ \max_{p} \min_{\delta \in R} E_{x \sim p}[\Phi(x) + \delta(x)]  \text{ eq. 10 } $$
where $R$ is the reward uncertainty set. It attributes the failure of GFNs to maximise $\Phi$ to the fact that the uncertainty set produced by its regulariser does not contain the proxy reward.

**Strengths:**

* The introduces an interpolated regulariser which unifies three different operators and a respective algorithm that achieves the desiderata of oversampling high reward, demonstrated by the empirical evaluation
* The paper characterises the uncertainty reward sets associated with the different regularisers and provides an explanation on why GFlowNets do not oversample high reward regions by relying on their equivalence entropy regularised policies: their reward uncertainty set does not include the proxy reward.

**Weaknesses:**

As I have said in a previous review of the same paper: The paper addresses the property optimization problem for (robust) sequence generation in an RL setting. I understand that the core of the paper is the robust property optimization with RL [...] however there are different approaches to the same problem that are [still] not discussed in the related work. In particular there is quite some work in generative models that seek to do exactly the same, i.e. property optimization.

See references provided in the previous review. I think it is still useful to discuss also such different modelling approaches to the same problem.

**Questions:**

* Robust optimization seeks to maximize an objective function in the worst case under uncertainties. The objective as given in equation (1) does not look like a robust optimization problem but rather takes $\Phi$ at face value and does not state anything about a worst-case setting. So why robust optimization is appropriate for eq. (1)?
$$ \max_{x_1, ..., x_k} \sum_i \exp{\Phi(x_i)} \text{ subject to } d(x_i, x_j) > \delta, \text{ eq. 1 } $$

* how should I understand equation 10:
$$ \max_{p} \min_{\delta \in R} E_{x \sim p}[\Phi(x) + \delta(x)], \text{ eq. 10 } $$
in the sense that $\delta$ is not something we have access to, is this more a theoretical concept to study the properties of the different regularisers? Reading it as is stated above there are two things that can be adjusted/learnt the sampling distribution $p$ and the $\delta$ function (or I guess equivalently the $\Phi$), however $\Phi$ is fixed. In addition the max/min problem reads as: find the distribution that maximizes $\Phi$ under the lowest error, this does not sound like optimization in the worst case setting.

---

> ### Author Response · Authors · 2025-11-21
>
> We would like to begin by thanking the reviewer for their thorough comments, helpful suggestions and questions. We’re glad the reviewer found that our algorithm **“achieves the desiderata of oversampling high reward, demonstrated by the empirical evaluation”**. We now hope to address their questions and comments below, and in an updated version of the paper (with changes highlighted in red).
>
> ---
> ### Missing discussion of generative modeling approaches.
> We would like to thank the reviewer for pointing out this oversight, for the references and apologize for the omission. We’ve since added a paragraph to the related work (currently in App. A due to lack of space) discussing the generative modeling approach to this problem.
>
> ---
> ### Interpretation of Eq. 10 and  $\delta$
> $\delta$ is indeed more of a theoretical concept. In essence, the idea is that there is a true reward function $r$, a proxy reward function $\Phi$ and $\delta$ is simply the difference between the two: $\delta(x):= r(x) - \Phi(x)$. We have no control over $\delta$, it is simply a way to model the discrepancy. As a result, solving Eq. 10 can be interpreted as finding the sampling distribution $p$ that maximizes our expected true reward $\Phi(x) + \delta(x)$ given the worst case possible $\delta$. In a sense, similarly to [1], $\delta$ can be thought of as an adversary trying to minimize our reward.
>
> To make the problem interesting, robust RL limits the possible values of $\delta$ to some feasible set $R$. Different shapes of $R$ correspond to different regularizers; we argue in this paper that the feasible set associated with TGM is a more realistic depiction of the uncertainty of the proxy reward as it allows for $\delta$ to be negative, whereas entropy regularization only allows for positive $\delta$.
>
> ---
> ### Why is robust optimization appropriate for Eq. 1?
> This is an excellent question. Eq. 1 is a formalization of the problem practitioners working on scientific discovery are actually trying to solve. Essentially, it boils down to a trade off between maximizing the reward of samples while ensuring they are diverse enough. Even though this problem is not directly equivalent to Eq. 10, we believe it is tightly linked to the latter as it also models the tradeoff of generating high proxy reward samples that are diverse.
> - Since we are maximizing $\Phi(x) + \delta(x)$, the robust method is encouraged to sample objects with high proxy reward.
> - The adversarial $\delta(x)$ ensures that our method samples diverse objects. Given the shape of the robust sets ($\delta(x)$ can reduce the reward of one object but not both), assigning too high probability to a single object allows $\delta(x)$ to reduce our expected reward substantially. To guard against the worst case, the optimal solution involves ensuring probability mass is spread on many objects => diverse sampling.
> - In particular, when using our compositional model of uncertainty $\delta(x) =\sum_{i=0}^n \delta_i[a_i]$ where uncertainty accumulates over a trajectory, the more similar 2 objects are, the more $\delta_i$ they tend to share. Hence, if we only generate similar samples (e.g. sequences who share the first $n$ tokens), the $\delta_i$ for these shared tokens can all be negative, leading to a low true expected reward.
> - Conversely, the more the sampled objects differ, the more tradeoffs $\delta(x)$ faces, eventually straining its “budget” such that at least certain objects must have high true reward. Thus, the robust policy is likely to sample more objects which would satisfy $d(x_i, x_j) > d_\mathrm{min}$. (We changed $\delta$ -> $d_\mathrm{min}$ in Eq. 1 to avoid confusion with the uncertainty notation).
>
> Ultimately, the goal of this model of scientific discovery is to find an object that maximizes the true reward while only having access to a proxy reward. If we correctly model our uncertainty regarding the proxy reward function and solve the corresponding robust problem, we can get guarantees over the quality of sampled objects.
>
> [1] Brekelmans, Rob, et al. "Your policy regularizer is secretly an adversary." arXiv preprint arXiv:2203.12592 (2022).

---

> > ### Author Response · Authors · 2025-11-27
> >
> > We hope our added discussion of generative modeling approaches in the related work is appropriate and that our response regarding the link between the robust/scientific discovery objectives was helpful. Thank you again for pointing out the omission, addressing it has improved the thoroughness of our related work. We'd be glad to discuss any other unaddressed concerns or any new questions!

---

### Official Review · Reviewer_QUN8 · 2025-11-01

**Soundness:** 3
**Presentation:** 2
**Contribution:** 3
**Rating:** 6
**Confidence:** 3

**Summary:**

This paper introduces trajectory general mellowmax, a reinforcement learning method for discrete compositional generation tasks like molecular or protein design. The authors highlight how methods that learn a policy whose density/distribution is proportional to the reward (such as GFlowNets) tend to produce diverse but generally low-quality samples (since the high reward, low density samples are drowned out by the low reward, high density samples). To address this, the authors propose the general mellowmax operator, which unifies and interpolates between existing soft RL operators to better balance diversity and quality. Experiments on synthetic and biological benchmarks show that the proposed approach consistently generates higher reward and still diverse candidates than prior methods.

**Strengths:**

This paper is original in unifying multiple soft RL operators through the proposed general mellowmax framework. The proposed trajectory general mellowmax is significant in bridging GFlowNets and robust RL under a common theoretical perspective. The paper contains rigorous mathematical derivations, a well motivated algorithmic design, and thorough experiments across both synthetic and real wold biological design tasks. The problem motivation, operator formulation, and empirical findings are clearly articulated, and the work is significant in meaningfully addressing a key limitation of excess diversity (at the expense of high reward designs) in GFlowNets / similar constructions.

**Weaknesses:**

The mathematical notation can feel a bit dense at times and I found myself getting occasionally lost on what the various key parameters are meant to govern. It would be helpful to have a better illustration or explanation on the role of alpha, q, and omega in TGM - this is attempted in Figure 3, but requires the reader to go to other portions of the paper to understand what the axes are describing. So in general, I feel that the presentation and clarity of the paper could be improved. I also feel like the experimental section would benefit from including a baseline that is also designed to address the mode concentration issues of GFlowNets, such as temperature conditioned GFNs.

**Questions:**

Are there ways to experimentally demonstrate the connection to robust RL? E.g., by introducing noise into the proxy reward to evaluate how TGM performs relative to GFlowNets?

Have you compared TGM against established alternatives that address policy over smoothness / diversity such as temperature conditioned GFlowNets? I saw temperature conditioned GFNs mentioned in the paper, but not used as a baseline for comparison.

For the biological sequence design experiments in 6.2, it would be useful to understand how the diversity of the generated sequences varies across the baselines and TGM with different choices of q.

---

> ### Author Response · Authors · 2025-11-21
>
> We would like to begin by thanking the reviewer for their time, feedback and suggestions. We’re glad the reviewer found our paper **“original”** and **“significant in bridging GFlowNets and robust RL”** and that the **“motivation [...] and empirical findings are clearly articulated”**. We now address their questions and comments below, and in an updated version of the paper (with changes highlighted in red).
>
> ----
> ### Dense mathematical notation
> We thank the reviewer for the useful feedback, we agree that the paper could benefit from a clearer description of the many hyperparameters used by TGM. To address this issue, we’ve added a section App. F.1  with a table describing TGM’s hyperparameters (we hope to include it in the main body once we have more space). We hope this change clarifies their use. We've also updated the caption of Fig. 2 and rewritten a part of Sec. 3 to hopefully make the exposition clearer.
>
> ----
> ### Comparison with temperature-conditioned GFlowNets.
>
> This is an excellent suggestion. We implement and test the baseline (with details and results in App. C.7. We follow the method described in [1] which expands on [2]. Specifically, during training, for each batch we uniformly randomly sample a temperature between the set minimum temperature and maximum temperature. This temperature is encoded by a two-layer MLP before being added to each transformer layer. Then, as in [1], the temperature is decoded into a scalar which is used as a temperature parameter, modulating the output logits.
>
> We then sweep over minimum temperatures $\\{1/256, 1/64, 1/16, 1/4 \\}$ and maximum temperatures $\\{1/128, 1/32, 1/8\\}$ as well as learning rates in $\\{1e-5, 1e-4, 1e-3\\}$. For evaluation, we sample using the minimum temperature. Overall, this leads to similar or slightly better performance (on UTR) compared to a regular GFN. Nonetheless, TGM still performs better overall.
>
> We also take the best performing hyperparameter setting from TGM and test it with temperature conditioning between $\frac{1}{\beta}$ and $\frac{2}{\beta}$. The change results in a significant performance increase for GFP **(2.95 -> 3.30)** showing that temperature conditioning is a general technique that can be applied to both GFNs and TGM.
>
> ----
> ### Evaluation of diversity
> We thank the reviewer for this suggestion. We have added a diversity metric described in [3] to App. C.3 to get a better understanding of how the diversity/novelty of generated samples differs between methods. Results and experimental details are included there.
>
> In terms of diversity, GFNs and TGM obtain similar values for UTR. For AMP, diversity is higher for GFNs but this is partially due to all the modes GFN (and TGM with $q=0$) generates being of length 60. Since the Levenshtein edit distance is not normalized for distance, it is expected that the pairwise distance will grow with sequence length. Conversely, the sequence lengths of modes generated by TGM are shorter, limiting the potential edit distance. For GFP, GFNs obtain noticeably higher diversity, though at the cost of significantly lower reward.
>
> Overall, the increase in average mode reward does come at the cost of reduced diversity, though we argue this tradeoff is fair given the goals of scientific discovery (there is still enough diversity to have $k$ different interesting candidates).
>
> ---
> ### Experimental evaluation of robustness
> For these tasks, to our knowledge, there is unfortunately no widely used benchmark for evaluating robustness. To test robustness, we would need to resort to a toy example where we artificially set the discrepancy between the proxy reward and the true reward. The results would heavily depend on how we set this discrepancy/noise.
>
> ----
> ### References
> [1] Kim, Minsu, et al. "Learning to scale logits for temperature-conditional gflownets." arXiv preprint arXiv:2310.02823 (2023).
>
> [2] Zhang, David W., et al. "Robust scheduling with gflownets." arXiv preprint arXiv:2302.05446 (2023).
>
> [3] Jain, M., Bengio, E., Hernandez-Garcia, A., Rector-Brooks, J., Dossou, B. F., Ekbote, C. A., ... & Bengio, Y. (2022, June). Biological sequence design with gflownets. In International Conference on Machine Learning (pp. 9786-9801). PMLR.

---

> > ### Author Response · Authors · 2025-11-27
> >
> > We hope our added experiments on temperature-conditioned GFNs/TGM, evaluations of diversity and table describing TGM hyperparameters were helpful. We're particularly grateful for the suggestions as we believe they have strengthened the contribution. Please let us know if you have any other unaddressed concerns or any new questions!

---

### Official Review · Reviewer_5kkr · 2025-11-01

**Soundness:** 3
**Presentation:** 3
**Contribution:** 3
**Rating:** 6
**Confidence:** 2

**Summary:**

The paper proposes Trajectory General Mellowmax, a novel algorithm that outperforms GFN in finding high-reward targets on discrete state spaces. The experiments conducted in both the synthetic environment and the biological sequence search demonstrate that the newly proposed methods are effective and relevant.

**Strengths:**

- The proposed methods are novel and theoretically grounded, with a clear motivation.
- The numerical experiments are comprehensive, showing the relevance of TGM

**Weaknesses:**

- The writing is pretty challenging to follow.
- The motivating examples of "high reward path dominated by many low reward paths" seem to be the issue of the designed target distribution rather than the problem of the learning algorithm. For a 0-1 reward, if we reduce the temperature of the distribution to be proportional to exp(r/gamma) where gamma -> 0, the valid target examples are only going to be concentrated on positive reward ones, and the cases presented in the motivating examples no longer seem to be valid.

**Questions:**

See weakness.

---

> ### Author Response · Authors · 2025-11-21
>
> We would like to begin by thanking the reviewer for their comments and time. In particular, we’re glad they found our method **“novel and theoretically grounded”** and that our **“numerical experiments are comprehensive”**. We now address their questions and comments below, and in an updated version of the paper (with changes highlighted in red).
>
> -----------
> ### Writing is challenging to follow
> We agree the writing can be challenging to follow as our paper brings together concepts from GFlowNets, RL, robust RL, etc. We’ve updated the main Sec. 3 to include a more thorough overview of the various operators, improved the caption of Fig. 2 and added Tab. 8 in the appendix summarizing all hyperparameters used by TGM. We are open to any other suggestions or comments on specific parts that were hard to follow!
>
> -----------
> ### Motivating example with 0-1 reward
> We believe this is an important point and thank the reviewer for raising it, as it pinpoints challenges that are specific to scientific discovery tasks. We agree that an ideal reward function might assign 0 reward to low quality samples leading to a sampling distribution that attributes probability only to high reward samples. Unfortunately, this is practically infeasible for the following reasons:
> - In many cases, we do not have access to the true reward function or a function that fits these criteria but to a proxy reward. This proxy reward, as illustrated in Fig. 2, is insufficiently discriminative between good and bad objects: almost all objects are assigned non-negligible rewards, fitting the assumption of the motivating example. Due to the exponential amount of such objects, sampling proportional to reward will lead to significant probability mass being diverted from high reward samples to low reward samples.
> - As pointed out by the reviewer, we can accentuate the discriminative power of the proxy by raising it to a large exponent.  We test this effect in Fig. 6 using  $\beta$ ($\frac{1}{\gamma}$ in the reviewer's comment). Although this trick theoretically works, the exponent values required to dwarf bad objects are problematic for the training process. Even with the VarGrad formulation of the loss and gradient clipping to stabilize training (necessary as the loss reaches the order of $\approx 10^6$ for $\beta=8192$), setting large $\beta$ for GFNs still does not reach the performance of TGM.
> - Finally, even if we had access to a 0-1 reward function that perfectly discriminates good and bad objects, such a reward function would introduce additional learning challenges. Initially, the generative model samples objects uniformly and thus would only generate bad samples with 0 reward that yield no training signal. In a sense, the continuity of rewards from bad to good is a useful signal for the model to learn to progressively generate better and better samples.

---

> > ### Author Response · Authors · 2025-11-27
> >
> > We hope our response was helpful in addressing the concern over the motivating example (with respect to the possibility of using a 0-1 reward). We'd be glad to discuss any other unaddressed concerns or new questions!

---

### Official Review · Reviewer_9b9M · 2025-11-01

**Soundness:** 4
**Presentation:** 4
**Contribution:** 2
**Rating:** 4
**Confidence:** 4

**Summary:**

The authors introduce the Trajectory General Mellowmax operator, motivated by the framework of regularized reinforcement learning operators, which tend to encourage more concentrated policy distributions, and seem to show improvements across datasets.

**Strengths:**

1. The paper addresses an important problem—developing a reinforcement learning method that can generate diverse responses while also producing high-scoring candidates useful for biological and chemical discovery.

2. The paper is well written and generally easy to follow.

**Weaknesses:**

1. It seems that several important baseline comparisons are missing (see questions for details).

2. I’m not sure it’s appropriate to compare GFN with TGM, since GFN involves training a Q-network and therefore requires significantly more computation. The compute budget should be made fair for a valid comparison (see questions for details).

3. I also find the motivating example somewhat unclear or unconvincing (see questions for details).

4. Some of the figures are hard to understand.

**Questions:**

1. It seems that several baselines ([A], [B], and [C]) are missing. Since these approaches also use a Q-function, comparing against them would make the evaluation fairer. In contrast, GFlowNets do not rely on a Q-function, which makes direct comparison more challenging, as their learning dynamics and compute requirements differ fundamentally. Moreover, in the middle panel of Figure 4, the GFlowNet performance appears to be increasing, does this suggest that GFlowNets could further improve with additional compute (e.g., equivalent to what is used for Q-function training)?

2. In the intuition for the motivating example, doesn’t the argument implicitly depend on the quality of the reward model? For instance, if the rewards were accurate, then each sequence wouldn’t necessarily need to have its reward lower bounded by a constant r, since poorly performing samples could simply receive large negative rewards. Wouldn’t this issue therefore be mitigated by using a more accurate reward model?

3. In general, GFlowNets aim to generate diverse and novel samples. With that in mind, could the paper report diversity and novelty metrics as discussed in [E] and [F]? Additionally, similar to Figure 5 in [D] and Figure 2 in [E], would it be possible to conduct synthetic experiments to evaluate whether TGM successfully recovers all modes?

4. Could you clarify how to interpret Figure 2? What exactly should the reader take away from it? (I don't think I understand the caption)

5. In lines 403–404, could you explain why $\delta = 28?$

---

**References**

[A] Mohammadpour, Sobhan, Emmanuel Bengio, Emma Frejinger, and Pierre-Luc Bacon. "Maximum entropy gflownets with soft q-learning." In International Conference on Artificial Intelligence and Statistics, pp. 2593-2601. PMLR, 2024.

[B] Lau, E., Lu, S., Pan, L., Precup, D., & Bengio, E. (2024). Qgfn: Controllable greediness with action values. Advances in neural information processing systems, 37, 81645-81676.

[C] Tiapkin, D., Morozov, N., Naumov, A., & Vetrov, D. P. (2024, April). Generative flow networks as entropy-regularized rl. In International Conference on Artificial Intelligence and Statistics (pp. 4213-4221). PMLR.

[D] Zhang, D., Malkin, N., Liu, Z., Volokhova, A., Courville, A., & Bengio, Y. (2022, June). Generative flow networks for discrete probabilistic modeling. In International Conference on Machine Learning (pp. 26412-26428). PMLR.

[E] Jain, M., Bengio, E., Hernandez-Garcia, A., Rector-Brooks, J., Dossou, B. F., Ekbote, C. A., ... & Bengio, Y. (2022, June). Biological sequence design with gflownets. In International Conference on Machine Learning (pp. 9786-9801). PMLR.

[F] Ekbote, C., Jain, M., Das, P., & Bengio, Y. (2022). Consistent training via energy-based gflownets for modeling discrete joint distributions. arXiv preprint arXiv:2211.00568.

---

> ### Author Response · Authors · 2025-11-21
>
> We would like to begin by thanking the reviewer for their thorough review and helpful feedback/suggestions. We’re glad they find that **“the paper addresses an important problem”** and that it is **“well written and generally easy to follow”**. We now address their comments and questions below, and in an updated version of the paper (with changes highlighted in red).
>
> -------
> ### Additional baselines
> We thank the reviewer for directing us towards these additional relevant baselines. We believe including them to strengthens our paper's experimental evidence. We implement [B] and [C] and discuss them below. The table of results is included in App. C.4.
> - [B] (QGFN), is a good idea that is complementary to ours (i.e. can also be used with TGM). Experimentally, we test the $p$-greedy QGFN setup (mixture of forward policy with argmax of Q-values). We train a separate Q network in parallel (using the same architecture as the GFN) and set $n$ (in the $n$-step return) to be the maximum sequence length/2. We sweep over learning rates $\\{1e-5, 1e-4, 1e-3\\}$, reward exponents in $\\{4, 16, 64, 256\\}$ and values of $p \in \\{ 0.25, 0.5, 0.75\\}$ while fixing the sampling temperature to 1. Overall, QTGM performs similarly to the best performing GFN hyperparameters.
>   - We also take the best performing hyperparameter setting from TGM and test it with the p-greedy setup (using p = 0.25). While it does not appear to improve performance, future combinations of these methods could be fruitful directions for future work (particularly leveraging the connection between the TGM policy and its associated Q-values).
>   - We have also added a discussion of [B] in our related work.
> - [C] We find that Munchausen DQN struggles to learn a good sampler. Similarly to SAC, the non-trajectory based loss appears to lead to unstable training and poor performance. We set $\alpha=0.1, l_0=-20$ and separate the entropy coefficient from $\alpha$ as we observe that entropy coefficients $<1$ tend to perform better (which would require $\alpha <0$). Like other methods, we run a hyperparameter sweep: over learning rates $\\{1e-5, 1e-4, 1e-3\\}$, reward exponents in $\\{4, 16, 64, 256\\}$ and entropy coefficients $ \\{1, 1/4, 1/16 \\}$ and select the best performing hyperparameters.
> - [A] since we focus on sequence generation tasks (i.e. trees) for which we do not need a backwards policy, the method of [A] will yield equivalent results to TGM with q=0 (which corresponds to path consistency learning). Investigating the performance of TGM for tasks corresponding to general graphs is beyond the scope of this paper.
>
> -------
> ### Computational requirements of TGM/Q-function
> We would like to politely push back against the assertion that TGM is more computationally expensive. TGM uses the exact same network as GFNs (with TB) in our experiments (i.e. a single transformer that takes in an input sequence and outputs logits) and is trained with the same style of trajectory loss. While the logits are technically Q-values (up to a scalar constant) in our case, they are directly linked to the policy by taking the softmax with inverse temperature $q\alpha + \omega$. Thus, both methods essentially just need to learn a single policy network (no additional Q-networks). In fact, our method only essentially only requires changing 2 lines of code in the loss compared to TB with VarGrad. As a result, the runtime/computational cost is essentially identical, as demonstrated in Tab. 4 in the appendix.
>
> -------
> ### Further compute for GFNs
> We run the experiment again for GFNs for an additional 800k samples (i.e. to reach 1 million samples) with 5 seeds. We include the average result below:
> | Samples | Modes Found | Avg Min Dist |
> |---------|-------------|--------------|
> | 0 | 0.0 | inf |
> | 100k | 40.6 | 26.81 |
> | 200k | 49.8 | 26.04 |
> | 300k | 54.0 | 25.59 |
> | 400k | 56.4 | 25.36 |
> | 500k | 58.0 | 25.20 |
> | 600k | 59.6 | 24.94 |
> | 700k | 59.8 | 24.78 |
> | 800k | 59.8 | 24.64 |
> | 900k | 59.8 | 24.47 |
> | 1M | 59.8 | 24.36 |
>
> GFNs do (eventually) end up finding all modes. However, given the nature of the task, it is expected that more modes will be found as we keep generating. Given the point above, we believe the comparison between GFNs and TGM in Fig. 4 is fair (i.e. they use very similar amounts of compute). Noticeably, the best GFN configuration, even with 5x more samples, still has a noticeably higher average minimum distance, demonstrating the superior performance of TGM.
>
> -------
> ### Use of $\delta=28$
> We follow [1] which proposes the task and uses $\delta=28$ as distance threshold ([B] also uses the same value). Since TGM begins to saturate the benchmark, we also include the average minimum distance as an alternative metric not based on a specific threshold. To compute the average minimum distance, we keep track of the distance of the closest generated sample to each mode over the course of training: the average minimum distance is just the mean of those distances.

---

> > ### Author Response · Authors · 2025-11-21
> >
> > (continued)
> >
> > ### Diversity/novelty metrics
> > We thank the reviewer for the suggestion of a more multi-faceted evaluation of performance.  We have added the metrics described in [E] in App. C.3 (including experimental details) to hopefully get a better understanding of how the diversity/novelty of generated samples differs between methods.
> > - Noticeably, in terms of diversity, GFNs and TGM obtain similar values for UTR. For AMP, diversity is higher for GFNs but this is partially due to all the modes that GFN (and TGM with $q=0$) generate being of length 60. Since the Levenshtein edit distance is not normalized for distance, it is expected that the pairwise distance will grow with sequence length. Conversely, the sequence lengths of modes generated by TGM are shorter, limiting the potential edit distance. For GFP, GFNs obtain noticeably higher diversity, though at the cost of significantly lower reward.
> > - For novelty, all methods are relatively similar for UTR and GFP. There is a noticeable difference for AMP, however this is likely due, once again, to the longer sequences being generated by GFNs.
> >
> > Overall, the increase in average mode reward does come at the cost of reduced diversity, though we argue this tradeoff is fair given the goals of scientific discovery (there is still enough diversity to have $k$ different interesting candidates).
> >
> > ----
> > ### Interpretation of Fig. 2
> > We apologize for the confusion; we’ve since updated the caption to hopefully make it clearer (any feedback would be appreciated)! The goal of the figure is to illustrate the distribution of proxy rewards in various biogen-related tasks. The main takeaway is that for all 3 domains, almost all samples have non-negligible proxy reward, demonstrating that the problem in the motivating example is realistic.
> > - To create the light blue histogram, we uniformly randomly generate 1 million sequences for each domain, which we then evaluate with the proxy reward function. The histogram is the distribution of resulting rewards.
> > - The dark blue histogram is the distribution of rewards of 1024 samples generated by TGM (best hyperparameters) at the end of training
> >   - Here, we illustrate how the modes sampled by TGM compare to the proxy reward of average objects. TGM is capable of sampling objects with significantly higher reward.
> > - The red line is the reward of the best object (as evaluated by the proxy reward) in the validation set of the dataset used to train the proxy reward.
> > ----
> > ### More accurate reward model
> > Indeed the motivating example does implicitly depend on the distribution of rewards of the reward function. With a good enough reward model, poorly performing samples would get no reward ensuring no probability mass is put on them by the optimal GFN. However, in practice there are several issues with this assumption.
> > - As discussed above and illustrated in Fig. 2, we often only have access to a proxy reward function which cannot perfectly discriminate the quality of samples and ends up assigning non-negligible rewards to essentially all of the exponentially many possible objects. In all three domains, almost all objects have a reward big enough to cause an issue (given exponentially many of them).
> > - Given this distribution, we can attempt to artificially modify it to lower the magnitude of low reward samples relative to high reward samples, for example by raising the rewards to some exponent. While theoretically this could work, our results in App. C.1 indicate that high values of exponents required are problematic for the training process. Even with the VarGrad formulation of the loss and gradient clipping helping to stabilize training (necessary as the loss can reach values $\approx 10^6$ for $\beta=8192$), increasing $\beta$ for GFNs still does not reach the performance of TGM.
> > - Finally, even if we had access to a model that perfectly discriminated good from bad samples (giving the latter no reward), it would likely be hard to use it to learn to generate good samples. When generating initially (essentially uniformly randomly), the model would only generate bad samples with 0 reward that yield no training signal. In a sense, we believe that the continuity of rewards from bad to good is key for the model to learn to progressively generate better and better samples.
> >
> > ----
> > ### References
> > [1] Malkin, Nikolay, et al. "Trajectory balance: Improved credit assignment in gflownets." https://arxiv.org/pdf/2201.13259 (2022).

---

> > ### Comment · Reviewer_9b9M · 2025-11-25
> >
> > I thank the authors for their detailed comments and ablations. I have one followup question, maybe I misunderstood this, but just to clarify the following point: *"We would like to politely push back against the assertion that TGM is more computationally expensive. TGM uses the exact same network as GFNs (with TB) in our experiments (i.e. a single transformer that takes in an input sequence and outputs logits) and is trained with the same style of trajectory loss. While the logits are technically Q-values (up to a scalar constant) in our case, they are directly linked to the policy by taking the softmax with inverse temperature $q\alpha + \omega$. Thus, both methods essentially just need to learn a single policy network (no additional Q-networks). In fact, our method only essentially only requires changing 2 lines of code in the loss compared to TB with VarGrad. As a result, the runtime/computational cost is essentially identical, as demonstrated in Tab. 4 in the appendix."*, am I right to understand that the Q function is derived from the policy network itself? Can some more details be provided for this?

---

> > > ### Author Response · Authors · 2025-11-26
> > >
> > > Essentially yes (up to some minor details)! The single network $Q_\theta$ we train is directly used as TGM policy. It is also linked to the TGM Q-values (similarly to trajectory balance GFN).
> > >
> > > **TB GFN**
> > >
> > > Typically, a network (let's denote it $F_\theta$) is trained that takes in states and outputs logits. The forward policy $P_F$ is then given by taking the softmax of $F_\theta$. At optimality, the logits of  $F_\theta$. are equal to the optimal log flows + some scalar (due to the properties of the softmax, we can only guarantee they are equal up to some scalar constant).
> > >
> > > **TGM**
> > >
> > > Analogously, in TGM, we train $Q_\theta$ that takes in states and outputs logits. The TGM policy is directly given by  $\mathrm{softmax}( (\alpha q +\omega) Q_\theta)$. At optimality, the logits of $Q_\theta$ are equal to the TGM Q-values, up to a scalar constant.
> > >
> > > We support this link theoretically in Theorem 3.1. In particular, the theorem states that, in a DCP, $\mathrm{softmax}( (\alpha q +\omega) Q_\theta)$ is the optimal policy of the regularized problem *if and only if* $Q_\theta$ satisfies the trajectory constraint on all trajectories in the DCP.

---

> ### Comment · Reviewer_9b9M · 2025-11-26
>
> Thanks for the clarifications and the additional experiments. I've updated the score accordingly.

---

### Author Response · Authors · 2025-12-03
**Final Comment for Area Chair**

We would like to thank the reviewers for their thoughtful and helpful feedback, as well as the AC for the extra work they will need to do given these unusual circumstances.

**Key points:**
- Two days before the leak, **reviewer 9b9M** had engaged with our rebuttal and raised their score substantially **(4 -> 8)**
- The paper had scores **[8, 6, 6, 6]** for an average of **6.5**.
- While other reviewer did not have the chance to engage, we believe we have addressed their feedback, concerns and questions thoroughly through our rebuttal.
  - In particular, there was noticeable overlap (reward functions, diversity, baselines) between the concerns of **reviewer 9b9M** (who increased their score substantially following our rebuttal) and **reviewers 5kkr** and **QUN8** (who didn't get to respond).

We summarize these points below.

----
## Strengths
Overall, reviewers appreciated the following strengths.

**Well-motivated + addresses an important problem**
- “[...] the work is significant in meaningfully addressing a key limitation of excess diversity [...]” (QUN8)
- “The paper addresses an important problem—developing a reinforcement learning method that can generate diverse responses while also producing high-scoring candidates [...]” (9b9M)

**Original/novel**
- “The proposed methods are novel [...]” (5kkr)
- “This paper is original in unifying multiple soft RL operators through the proposed general mellowmax framework” (QUN8)

**Theoretically grounded**
- “[...] provides an explanation on why GFlowNets do not oversample high reward regions by relying on their equivalence entropy regularised policies: their reward uncertainty set does not include the proxy reward.” (iNss)
- “The proposed methods are [...] and theoretically grounded, with a clear motivation.” (5kkr)
- “The proposed trajectory general mellowmax is significant in bridging GFlowNets and robust RL under a common theoretical perspective. The paper contains rigorous mathematical derivations [...]” (QUN8)

**Experimentally rigorous**
- “The numerical experiments are comprehensive, showing the relevance of TGM.” (5kkr)
- “[...] thorough experiments across both synthetic and real wold biological design tasks.“ (QUN8)
- “[...] achieves the desiderata of oversampling high reward, demonstrated by the empirical evaluation.” (iNss)

**Well-written**
- “The paper is well written and generally easy to follow.” (9b9M)
- “The problem motivation, operator formulation, and empirical findings are clearly articulated [...]” (QUN8)

---
## Improvements
The reviewers mentioned the following main weaknesses and feedback. We’ve made the improvements described below to the paper to address these concerns (highlighted in red).

**1. Missing baselines (Reviewer 9b9M, QUN8):**  Could be strengthened by comparing TGM with MunchausenDQN, QGFN, and temperature-conditioned GFNs.

We implemented and tested all three methods. The latter two are complementary to the TGM operator. To illustrate this, we test their techniques with TGM as well.
- MunchausenDQN performs poorly. QGFN and temperature-conditioned GFNs sometimes improve performance over GFNs. The best performing method remains either TGM or TGM with temperature-conditioning (yields a noticeable improvement for the GFP task, 2.95 -> 3.3).

|       | M-DQN | GFN | QGFN | GFN + Temp | Best TGM | QTGM | Best TGM + Temp |
|-------|-------|-----|------|------------|----------|------|-----------------|
| UTR   | 3.37±0.12 | 4.12±0.00 | 4.08±0.04 | 4.18±0.05 | **4.27±0.01** | 4.05±0.06 | 4.18±0.01 |
| AMP   | 2.97±0.82 | 10.04±0.04 | 8.96±0.44 | 9.97±0.12 | **10.43±0.07** | 9.41±0.25 | 9.83±0.34 |
| GFP   | - | 1.90±0.03 | 1.08±0.07 | 1.89±0.04 | 2.95±0.08 | 0.99±0.14 | **3.30±0.15** |

**2. Evaluation of diversity/novelty (Reviewer 9b9M, QUN8):**  Requested an evaluation of the sampling diversity and novelty of TGM relative to other methods.

We test the metrics described in [1] with the Levenshtein edit distance. Ultimately, TGM does tend towards less diversity/novelty than GFN in some domains, but remains far from exhibiting a collapse in diversity. The tradeoff between quality and diversity is expected, and we believe it is more reflective of the goals of scientific discovery. The results can be found in Tab. 4 and Tab. 5 in the appendix.

**3. Clarity of writing (Reviewer 9b9M, 5kkr, QUN8):**  Found that the paper could be challenging to follow, a caption was unclear, and that the notation was dense at times.

To remedy this (changes highlighted in red), we add a more thorough discussion of the various soft RL operators, modify the confusing caption and add a table describing the hyperparameters used by TGM.

**4. Missing related work (Reviewer iNss):** Mentioned we were missing a discussion of generative/diffusion model approaches to scientific discovery.

We’ve added a section in the related work discussing the generative modeling approaches to a similar problem and contrasting them to the approach taken by GFNs/TGM.

---

> ### Author Response · Authors · 2025-12-03
> **Final Comment for Area Chair (continued)**
>
> ## Questions/Concerns
> Finally, we answered their main questions/concerns as follows.
>
> **Reward function (Reviewer 9b9M, 5kkr):** Reviewers questioned whether the motivating example in the paper was really an issue with the GFN objective or just the reward function itself.
>
> An ideal reward function could perfectly discriminate between poor-quality samples and high-quality samples, assigning 0 reward to the former. As a result, an exponential amount of poor-quality samples would not dilute the probability mass assigned to the modes of the distribution.
> We argue that while this would hold in theory, there are multiple issues with such an assumption, which we detail in our response. To summarize:
> - In practice, proxy rewards assign non-negligible rewards to almost all samples (as illustrated in Fig. 2).
> - We test inducing this separation by raising the reward exponent. Even with tricks to stabilize performance for GFNs, this still yields worse performance than TGM (as illustrated in Fig. 6).
> - Even if we had access to such a perfect reward function, it could cause issues by not providing enough training signal for the algorithm.
>
> > **Reviewer 9b9M** seemed convinced by these arguments as they raised their score from **4 -> 8**. The discussion period was cut off before **reviewer 5kkr** could engage.
>
> **Computational requirements (Reviewer 9b9M)** The reviewer questioned the computational complexity of TGM, as training an additional Q-network can be expensive.
>
> We clarified that TGM trains a single network (like trajectory balance GFNs) which is directly linked to the policy. As such, the computational complexity is essentially identical, as illustrated in Tab. 3.
> > **Reviewer 9b9M** appreciated the clarification
>
> **Link between robust equations and scientific discovery (Reviewer iNss):** The reviewer was interested in how the robust RL equation related to the goal of scientific discovery.
>
> We clarify that we are indeed solving a robust RL equation where we want to maximize reward with respect to a worst-case deviation from the proxy reward. In particular, while the two objectives aren’t strictly equivalent, the structure of the robust sets induced by our compositional form of uncertainty encourages a diverse sampling distribution, matching the requirements of scientific discovery.
> > The discussion period was cut off before **reviewer iNss** could engage.
>
> ---
>
> ## Conclusion
> Ultimately, while not all other reviewers had the chance to respond, we hope to have addressed most (if not all) of their concerns/questions. This feedback was helpful and led to a noticeable improvement in the paper and for reviewer 9b9M to raise their score **(4 -> 8)**. While, unfortunately, we have no way to prove no collusion took place, we hope the summary above provides ample evidence that scores of **[8, 6, 6, 6]** (or higher) would have resulted from the rebuttal under normal circumstances.
>
>
>
> [1] Jain, Moksh, et al. "Biological sequence design with gflownets". ICML 2022.

---

### Meta-Review · Area_Chair_tq13 · 2026-01-05

**Summary:**

Strength: The problem in consideration is timely and relevant. The presentation is also appropriate.

Weakness: The related work section in the literature could be improved and some of the mathematical content e.g., notations, could be better presented.

**Reviewer Concerns:**

Weakness: The related work section in the literature could be improved and some of the mathematical content e.g., notations, could be better presented.

**Reviewer Scores:**

The authors provided the information that the reviewer giving a "4" updated the score to "8". The other reviewers did not change their scores.

---

### Decision · Program_Chairs · 2026-01-26

Accept (Poster)